# Sequence learning recodes cortical representations instead of strengthening initial ones

Kristjan Kalm ●*, Dennis Norris

MRC Cognition and Brain Sciences Unit, University of Cambridge, Cambridge, United Kingdom

* kristjan.kalm@gmail.com

**Data Availability Statement:** Data is fully available at the 'Apollo - University of Cambridge Repository'. The link for the data is https://www.repository.cam.ac.uk/handle/1810/321717, or the DOI format: https://doi.org/10.17863/CAM.68802'.

## Abstract

We contrast two computational models of sequence learning. The associative learner posits that learning proceeds by strengthening existing association weights. Alternatively, recoding posits that learning creates new and more efficient representations of the learned sequences. Importantly, both models propose that humans act as optimal learners but capture different statistics of the stimuli in their internal model. Furthermore, these models make dissociable predictions as to how learning changes the neural representation of sequences. We tested these predictions by using fMRI to extract neural activity patterns from the dorsal visual processing stream during a sequence recall task. We observed that only the recoding account can explain the similarity of neural activity patterns, suggesting that participants recode the learned sequences using chunks. We show that associative learning can theoretically store only very limited number of overlapping sequences, such as common in ecological working memory tasks, and hence an efficient learner should recode initial sequence representations.

## Author summary

The ability to remember multiple individual events as a sequence is necessary for most complex human tasks. There is clear evidence that human sequence learning is accompanied by change in the way sequences are represented in the brain but the exact nature of the change remains unclear. In this study we use brain imaging to ask what is the neural mechanism underpinning sequence learning: we contrast two computational models of learning—associative and recoding—and test their predictions with neural activity data from the dorsal visual processing stream. We provide evidence that, instead of strengthening the initial cortical representations of sequences, learning proceeds by recoding the initial stimuli using a different set of codes. Furthermore, we show that associative learning without recoding is not theoretically capable of supporting long-term memory of short ecological sequences present in every day tasks such as reading, speaking, or navigating.

The scripts required to replicate the analysis of the fMRI data and all figures and tables presented in this paper are available at https://gitlab.com/kristjankalm/fmri_seq_ltm.

**Funding:** This research was supported by the Medical Research Council UK (DN, SUAG/012/RG91365, https://mrc.ukri.org/). The funders had no role in study design, data collection and analysis, decision to publish, or preparation of the manuscript.

**Competing interests:** The authors have declared that no competing interests exist.

## Introduction

Here we investigate the neural mechanism involved in learning short visual sequences. The ability to remember or to perform events or actions in the correct order is critical to the performance of almost all cognitive tasks [1]. Understanding human sequence learning mechanism is crucial not only for understanding normal cognition, but also to understand the nature of the impairments and disabilities that follow when sequence learning is disrupted [2–4].

In this study we ask whether the changes in neural activity during sequence learning reflect a particular type of optimal learning strategy. An optimal learner is an agent whose internal model reflects the statistics of the environment [5, 6], and human learning has been shown to follow the optimal model in a wide range of domains such as speech and language [7, 8], visual scenes and objects [9–13], and sensorimotor control [14, 15]. However, statistical regularities across sequences can be represented in multiple ways [1]. First, sequences can be represented as simple associations (Fig 1A and 1B) and their statistics represented by weighting the associations based on their relative frequency (Fig 1A–1C). An optimal learner would update the association weights as new data comes in to reflect the statistics of the environment. Alternatively, learning can proceed by recoding frequently occurring associations using new latent representations. The latter approach has been termed 'chunking' in cognitive literature [16, 17] to describe learning where complex objects (words, faces) are constructed from lower-level features (phonemes, syllables, oriented lines, [10]). The crucial difference between these two learning approaches is that the dimensionality of sequence representations changes: for associative learning the sequence representations remain the same, whilst new codes are inferred with recoding (Fig 1D). Therefore, we can dissociate between these two mechanisms by comparing neural representations of novel sequences to learned ones.

Research on sequence learning has provided evidence for both learning mechanisms. Manual motor skill learning has been shown to decrease noise in learned representations [18–20] whilst not changing the representations of individual items in the sequence [21, 22]. Similarly, in the auditory domain frequently co-occurring sequence items elicit a neural response that indicates an increase in association strength [23]. Contrastingly, chunking has been observed widely in tasks where separate movements are integrated into a unified sequence [24], and in auditory-verbal sequence learning [25–27], where multiple co-occurring sequence items are bound together and recalled in all-or-nothing fashion [28, 29].

Importantly, both learning mechanisms reduce the amount information required to represent stimuli [5, 11, 30] and therefore are hard to dissociate on the basis of simple univariate learning measures. For example, several past fMRI learning studies have observed two broad effects for learned stimuli: reduction of the BOLD signal and increase in pattern separability [18, 19, 31–34]. However, such results do not inform us of the computations underpinning the learning process: any statistical learning mechanism will reduce uncertainty and hence decrease resource requirements for learned stimuli [5, 35]. Therefore broad univariate measures indicating more efficient coding of learned stimuli, such as improvement in behavioural performance, reduction in the average BOLD response, or pattern separability, are expected *a priori* for any learning mechanism. Contrastingly, in this study we use fMRI to ask what is the computational mechanism underpinning learning in our task, rather than where in the brain can we detect learning effects.

We first formally derive the associative and recoding models in the context of Bayesian optimal learner. We show that the two accounts make dissociable predictions as to how sequences are encoded in the brain, and these predictions can be expressed in terms of the similarity of neural pattern activity. We tested these predictions in the dorsal visual processing stream using a sequence recall task together with the representational similarity analysis (RSA,

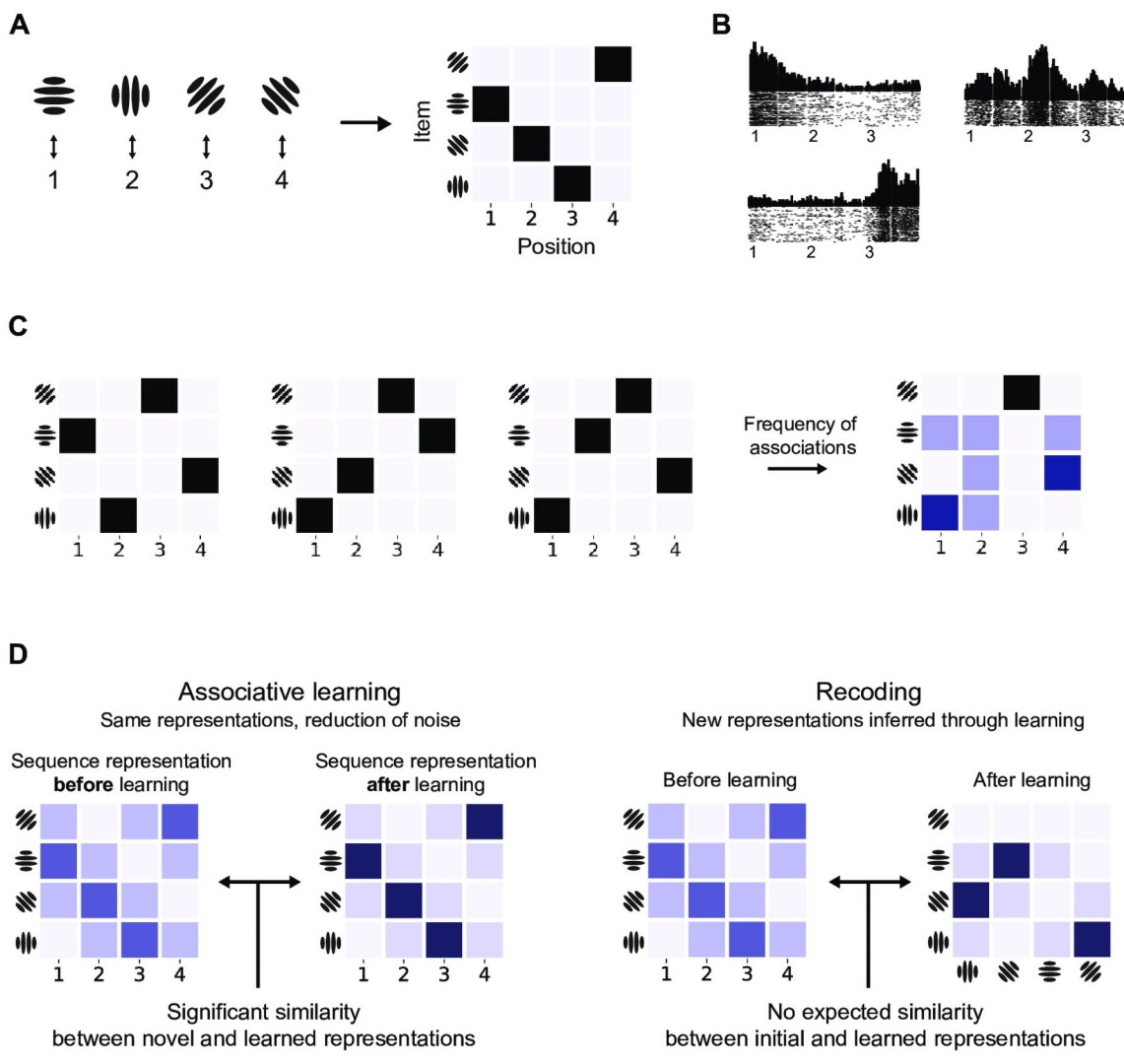

**Fig 1. Sequence learning.** (A) Four Gabor patches (items used in this study) associated with four sequence positions and a matrix representation of the sequence. (B) Item-position associations in monkey prefrontal cortex as observed by Berdyyeva et al. [38]. Each subplot displays spiking activity for a particular neuron: the first one responds most to items at the beginning of a three-item sequence, the second for the ones in the middle, and the last for items at the end of the sequence. Numbers on x-axis mark the onset of the stimulus events. (C) Visual representation of three sequences as position-item associations and the resulting frequency of associations. The frequency of associations can be learned as a model of the environment. (D) Dissociating between learning mechanisms in terms of similarity between novel and learned sequences: with associative learning (left) learned sequences share the same item codes with novel ones. Furthermore, learning reduces noise in learned sequence representations. Recoding (right) changes item representations so that novel and learned stimuli do not share representations.

[36, 37]) of fMRI data. We observed that only the recoding account can explain the similarity of neural activity patterns. Specifically, the code for sequences changed from representing novel sequences as individual items to representing them as chunks after they had been presented several times.

Finally, we show that associative learning can effectively store only a limited number of similar (overlapping) sequences. Therefore an efficient learner should benefit from recoding initial sequence representations, since ecological learning tasks, such as reading or navigating, often involve a large number of multiple overlapping sequences (e.g. words, directions, recipes).

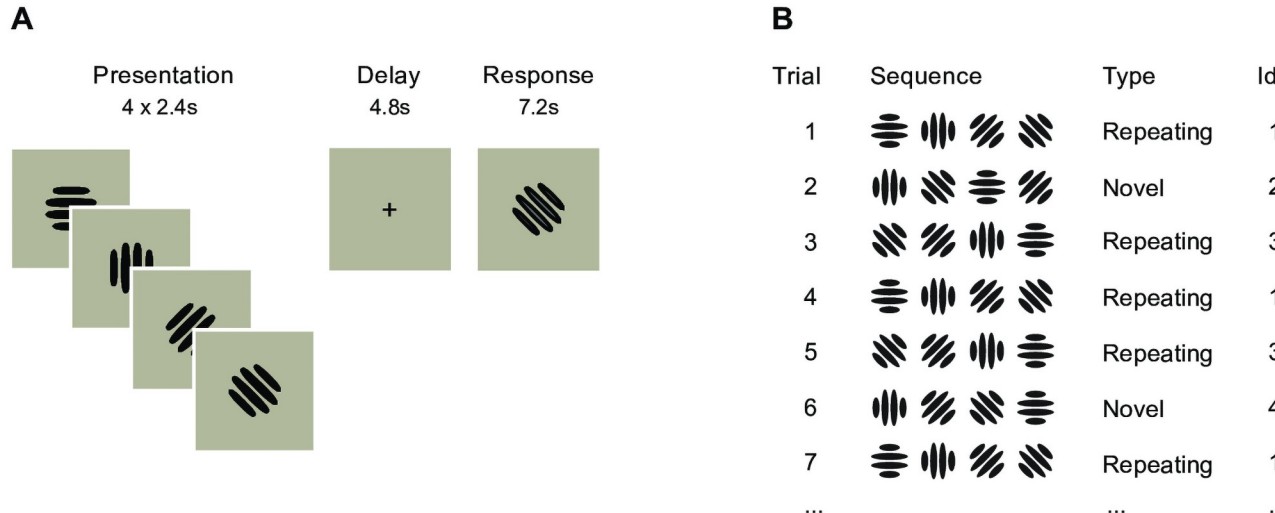

**Fig 2. Task.** (A) Single trial: participants had to recall a sequence of four Gabor patches in the order they were presented after a 4.8s delay period using a button-box. The size of the stimuli within the display area is exaggerated for illustrative purposes. (B) Trial types and progression: 2/3 of the trials were repetitions of the same two individual sequences (*repeating* sequences), while 1/3 of the trials were novel unseen orderings of the items (*novel* sequences). The identity and order of repeating and novel sequences were pseudo-randomised across participants.

Taken together our findings represent strong theoretical and empirical evidence for a specific learning mechanism: human learning of short visual sequences proceeds by recoding initial sequence representations with new ones within the same brain regions. Such recoding is necessary to enable efficient behaviour in complex tasks.

## Models

To contrast the predictions of the two learning models we used a simple sequence recall task: participants were asked to recall a sequence of four Gabor patches in the order they were presented after a brief delay period using a button-box. During the experiment some of the sequences were presented only once (*novel* sequences) and some were presented multiple times (*repeating* sequences, Fig 2). Only two individual sequences were repeated and we first presented them 12 times each during a practise session. This ensured that those two individual sequences were learned to criterion before the beginning of the main experiment. The repeating and novel sequences were designed maximally dissimilar to each other so that learning of the repeating sequences would not transfer to the novel ones. We proceeded to present the two familiar repeating sequences interleaved with novel sequences (Fig 2B).

### Sequence learning models

Here we define two sequence learning mechanisms—associative and recoding– in terms of two components: a model of representation for novel sequences and another for learned sequences. We assume that the difference between these two representations is attributable to the effect of the learning mechanism. Specifically, associative and recoding mechanisms make different predictions about the similarity between novel and repeating sequences. These predictions are formalised as representational dissimilarity matrices (RDM, Fig 3), which are then compared to the fMRI activity patterns using the representational similarity analysis (RSA, [36], Fig 3).

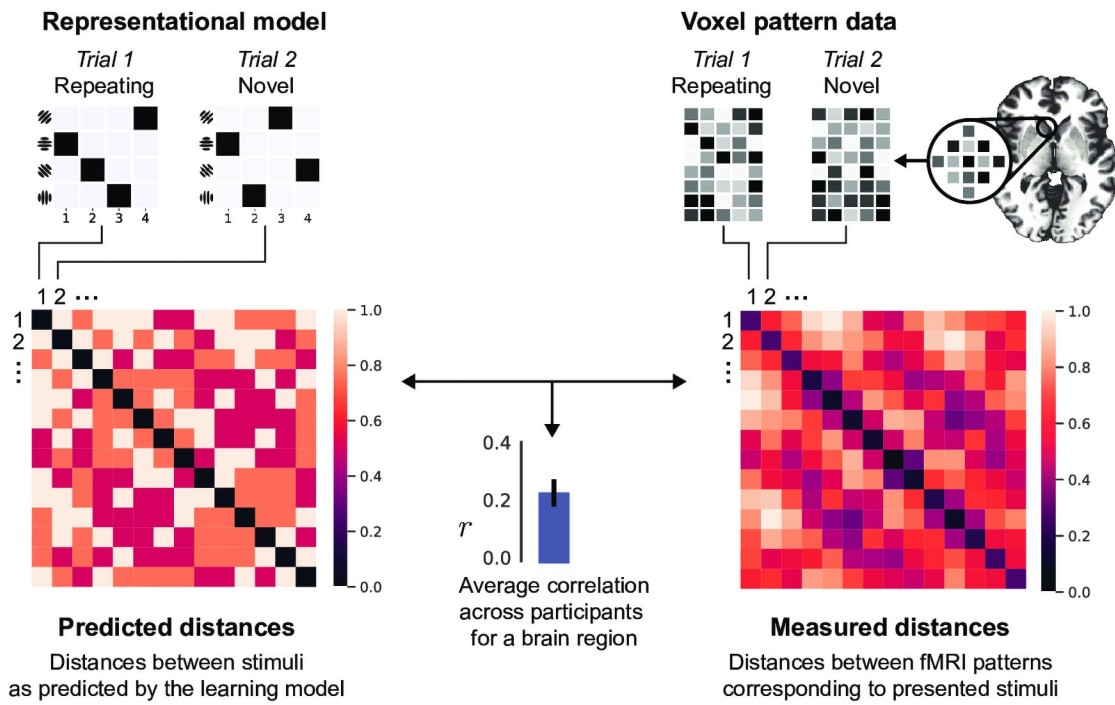

**Fig 3. Testing the predictions of learning models using RSA.** *Left*: model prediction expressed as a representational dissimilarity matrix (RDM) of pairwise between-stimulus distances (the cells in the matrix display the associative learner predictions in terms of item-position associations as quantified by the Hamming distance). The small matrices on the top refer to the representations of individual sequences in the matrix form (as shown on Fig 1). For example, second cell in the first row is the predicted Hamming distance between sequences presented on trials 1 and 2. *Right*: RDM of measured voxel activity patterns elicited by the stimuli. The small matrices are illustrative representations of voxel patterns from an arbitrary brain region. The correlation between these two RDMs reflects the evidence for the predictive model. The significance of the correlation can be evaluated via permuting the labels of the matrices and thus deriving the null-distribution. See Representational similarity analysis (RSA) in Methods for details.

**Associative learning model.** An ideal learner should infer internal representations that reflect the statistics of the environment. Intuitively, associative learning can be thought of as changing the weights of associations so that they reflect the frequency of past occurrences (Fig 1C). This can be formalised for sequences using the Dirichlet-Multinomial distribution, which encodes how many times particular discrete associations have occurred: the full description of the model and its parameters can be found in Associative learning in Methods.

The associative learning model makes two predictions that can expressed in terms of similarity between novel and repeated sequences (Fig 3). First, both sets of sequences should be encoded with the same sequence representation model since associative learning only changes the noise levels in the representations. In other words, if novel sequences are encoded as item-position associations, so should the learned ones. We tested this hypothesis with two classical sequence representation models: item-position associations, where sequences are formed by mapping items to their ordinal positions; and item-item associations, where consecutive items are associated with each other (see Sequence representation models in Methods).

Second, the associative learner predicts that the repeating (learned) sequences should be represented with less noise: the repetition of sequences should strengthen the weights of individual associations. Therefore, noise in the neural activity patterns generated by novel sequences should be greater than for repeating sequences and hence the expected similarity

*between* repeating and novel sequences should be greater than *within* novel sequences (see Associative learning predictions for RSA in Methods).

**Recoding model.** The recoding model posits that statistical regularities across sequences can be used to infer representations where frequently co-occurring stimuli are recoded using a single code. For example, if two individual items in a sequence occur next to each more frequently than apart, then an optimal learner should infer a model of the environment where those two adjacent items have been generated by a single latent variable. Formally, participants' internal representations of sequences are therefore recoded as latent variables given the observed sequences:

$$p(\theta|\mathbf{S}) = \frac{p(\mathbf{S}|\theta)p(\theta)}{p(\mathbf{S})}, \tag{1}$$

where $\theta$ is the internal latent model of a set of sequences $\mathbf{S} = \{\mathbf{y}_1, \ldots, \mathbf{y}_m\}$. Here we call this latent representation a *chunking model*, in line with previous literature [1, 16, 17]. The similarity between individual sequences is here defined by the inferred chunks, and not in terms of 'initial' items and their positions.

A chunking model $\theta_i$ is defined by two parameters and their probability distributions $p(\mathbf{x}, \mathbf{z}|\theta_i)$, where $\mathbf{x}$ is a set of individual chunks, and $\mathbf{z}$ a set of mappings defining how chunks are arranged together to encode observed sequences. For example, regularities within a set of two sequences $\mathbf{S} = \{ABCD, CDAB\}$ can be expressed by two chunks $\mathbf{x} = \{AB, CD\}$ and their mappings to the observed data $\mathbf{z} = \{((A, B), (C, D)), ((C, D), (A, B))\}$. Here we represent chunks formally as *n-grams*: for example, a four-item sequence *ABCD* can be represented by a tri-gram *ABC* and a uni-gram *D*; or two uni-grams *A* and *B* and a bi-gram *CD*, etc.

Next, we estimated the optimal chunking model for the sequences in our task: given the many possible ways sequences could be chunked, we assumed that the optimal learner would employ a chunking model that finds the most efficient encoding.

The full formal description of the chunking models, their parameters, and the process of inferring the optimal model can be found in Chunk Learning in Methods. Importantly, we designed the presentation of repeating and novel sequences so that the optimal model would remain the same for every trial across the experiment: every repeating sequence was encoded with a single four-gram chunk, and every novel sequence with four uni-grams (Fig 4, bottom row). Knowing the optimal chunk representation allowed us to calculate pairwise distances between sequences as defined by their constituent chunks. The resulting RDM of n-gram distances was then compared to the neural activity patterns using the RSA method (Fig 3).

Note that the optimal chunking model predicts the same representation for novel sequences as the associative item-position model. This is because the optimal chunking model encodes novel sequences with four one-item chunks resulting in the same number of item codes associated with the same positions (see Fig 4). In other words, both models' predictions for novel sequence representation are the same. However, the two models make different predictions about the similarity *between* the repeating and novel sequences.

## Results

### Behaviour

We observed that novel and repeating sequences were processed differently by participants. We calculated two behavioural measures of recall for both types of sequences: how many items were recalled in the correct position, and the average time between consecutive key presses. The proportion of correctly recalled items was roughly the same for novel and repeating sequences: 0.96 vs. 0.97, with no significant difference across participants ($p = 0.22$, $df = 21$).

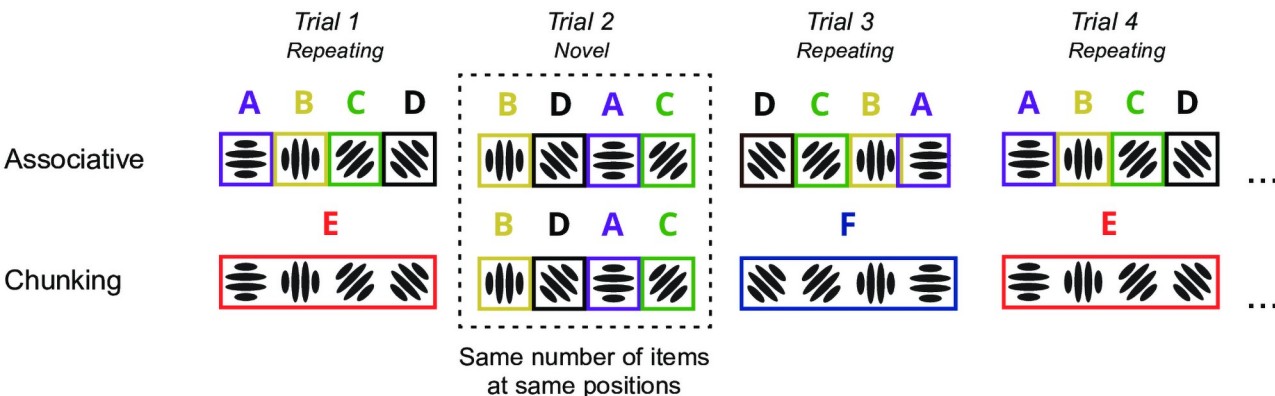

**Fig 4. Sequence representation in associative and recoding models.** The figure illustrates difference in sequence representations for four individual sequences: associative representations are displayed at the top and chunk codes at the bottom row. Differently coloured letters and boxes refer to individual item codes. For the chunk recoding model (bottom) item codes reflect the optimal chunking structure for the sequences presented in our experiment. Note that the representation of the novel sequence (Trial 2) contains the same number of item codes at same positions for both models.

This was expected since both novel and repeating sequences were only four items long and should therefore fit within participants' short term memory spans. However, participants were consistently faster in recalling repeating sequences: the average time between consecutive key presses was 0.018 seconds shorter for repeating sequences (0.534 vs. 0.552 sec, $t = -3.04$, $p = 0.007$, $df = 21$).

Next, we sought to establish how the neural representation of novel sequences differs from the repeating, learned ones: specifically, whether there is a change in representation that supports either the associative learning or recoding hypotheses.

### fMRI evidence for learning models

First, we parcellated the dorsal visual processing stream bilaterally into 74 anatomically distinct regions of interest (ROI). Here we focussed on the dorsal visual pathway, which covered the expected anatomical localisation of our task variables from visual presentation to manual recall (see Discussion for further details). In every ROI we compared the predictions of the sequence learning models to the voxel activation patterns for all three task phases (presentation, delay, response).

*Only the recoding model predicts the similarity between novel and repeating sequences*

We found significant evidence for the recoding model in three brain regions: the parietal inferior-supramarginal gyrus, the postcentral sulcus, and the occipital superior transversal sulcus (Fig 5). As predicted by the recoding model, the representation of sequences in all three regions followed a model where novel sequences are encoded with four one-item chunks but repeating sequences with single chunks, indicating a change in the representational code. The evidence for the recoding model was only statistically significant for the presentation phase of the task and not during the delay or the response phases.

*No evidence for associative learning in neural representations*

We found no evidence for the associative learning predictions: novel and repeating sequences were not encoded similarly in any of the brain regions. To further explore this null-result, we looked at the representation of novel and repeating sequences separately. We found significant evidence for novel sequence representation in eleven regions in the dorsal visual processing stream (Table 1; see also S1 Text for plots for individual brain regions). However,

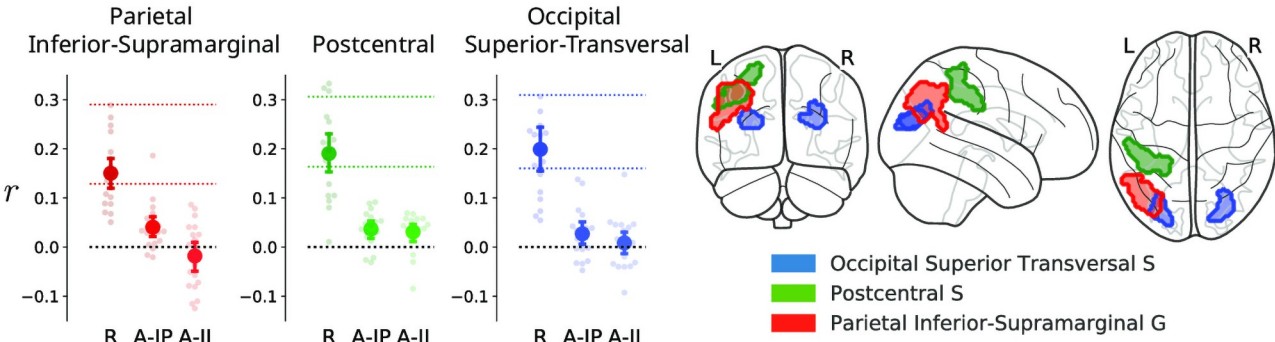

**Fig 5. Evidence for the recoding model.** The recoding model predicted the distance between pairs of voxel activity patterns corresponding to novel and repeating sequences in three brain regions. Models are shown on the x-axis ('R': recoding, 'A-IP' and 'A-II' are associative item-position and item-item models respectively). Y-axis displays the model fit in terms of participants' average Spearman's rank-order correlation (*r*). Dots represent individual participants' values and error bars the standard error of the mean (SEM). Coloured dashed lines represent the lower and upper bounds of the noise ceiling for the recoding model. In all displayed plots the lower noise ceilings were significantly greater than zero across participants. The anatomical contours of the regions are superimposed on the MNI152 glass-brain template (left).

in all of the regions where the associative model predicted similarity between novel sequences, it failed to predict the similarity between novel and repeating sequences ($df = 21$, $p > 10^{-3}$). This shows that, contrary to the predictions of the associative models, repeating and novel sequences did not share a common representational code in our task.

The associative learning model also predicts that the noise in the activity patterns generated by novel sequences should be greater than for repeating sequences and hence the expected similarity *between* repeating and novel sequences should be greater than *within* novel sequences. In other words, it should be easier to find evidence for associative codes between repeating and novel sequences than for novel sequences alone. Hence the lack of evidence we observe for associative learning cannot be attributed to the lack of fMRI measurement sensitivity.

*No behavioural evidence for associative learning*

There was also no behavioural evidence for associative learning: increased probability for associations present in the repeating sequences should affect novel sequences where such

**Table 1. Representation of novel sequences.** Anatomical region suffixes indicate gyrus (G) or sulcus (S). Asterisks (*) represent significant evidence for the item-position sequence representation model reaching the lower bound of the noise ceiling in any of the three task phases: presentation, delay, and response. The lower noise ceilings were significantly greater than zero for all regions displayed in the table ($df = 21$, $p < 10^{-3}$; see see Noise ceiling estimation in Methods for details).

| Name | Lobe | Presentation | Delay | Response |
|---|---|---|---|---|
| Central S | Frontal | | * | |
| Cuneus G | Occipital | | | * |
| Occipital Superior G | Occipital | | | * |
| Occipital Superior Transversal S | Occipital | | | * |
| Paracentral G S | Frontal | | * | * |
| Parietal Inferior-Supramarginal G | Parietal | * | | |
| Parietal Superior G | Parietal | | * | |
| Postcentral G | Parietal | * | | |
| Postcentral S | Parietal | * | | * |
| Precuneus G | Parietal | | | * |
| Temporal Superior S | Temporal | | * | |

associations are also present. For example, repeated exposure to a sequence *ABCD* should also boost *BDCA* since *C* appears at the 3rd position in both. We tested this prediction by comparing response times for individual item-position associations in novel sequences: there was no advantage for those individual associations which were shared with the two repeating sequences ($t = 0.28$, $p = 0.78$, $df = 21$).

## Model-free fMRI analyses of learning effects

We carried out two additional model-free fRMI analyses contrasting the representation of novel sequences to repeating ones. This was done to gauge how consistent our results were with previous fMRI studies which have shown two broad fMRI learning effects: reduction of the BOLD signal and increase in fMRI pattern separability for learned stimuli [31–34].

**Univariate BOLD difference for learned stimuli.** We carried out a whole-brain univariate analysis to test whether the average BOLD response differed between novel and repeating sequences. We found extensive bilateral reduction in the mean BOLD response for repeating sequences (Fig 6). This extended across parietal and pre-motor regions and was mostly absent in the primary visual and motor areas.

Note that the univariate change for the repeating sequences does not address the main hypothesis of this study, neither does it provide an alternative explanation of the data. Any neural learning mechanism is expected to make representations more efficient and therefore decrease the computational and metabolic cost of inference [5, 35]. Both associative learning and recoding predict more efficient representations: we cannot dissociate between retaining the same codes (associative learner) and recoding by simply measuring behavioural improvement or total change in metabolic cost (univariate BOLD).

*Changes in voxel pattern noise*

To gain more insight into learning-induced changes we tested whether the voxel pattern distances within and between novel and repeating sequences change across the experiment. For example, do the neural voxel patterns corresponding to the two repeating sequences become more dissociable over the experiment? Specifically, we tested for significant changes in voxel pattern distance (a) between the repeating sequences, (b) within the individual repeating sequences, (c) between the repeating and novel sequences, (d) within novel sequences. For full details on the distance analyses see Model-free fMRI analyses of learning effects in Methods. We found no brain regions where any of the voxel pattern distance change measures were statistically significant across the participants ($df = 21$, $p > 10^{-3}$).

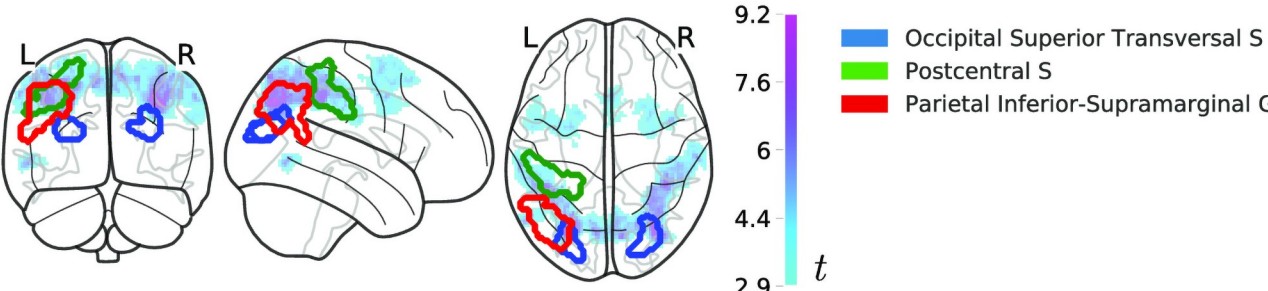

**Fig 6. Univariate effects of learning.** Statistical map of *t*-values (magenta-cyan) of the univariate BOLD difference for learned stimuli (repeating/learned < novel sequences) superimposed on the MNI152 glass-brain template. Regions which encode both novel and repeating sequences as predicted by the recoding model are projected on top of the statistical image with solid lines (red: the parietal inferior-supramarginal gyrus; green: the postcentral sulcus; blue: the occipital superior transversal sulcus).

Note that these model-free analyses were fundamentally different from the RSA approach employed for the comparison of the learning models above: here we did not measure the change in distances as predicted by a learning model but instead gauged whether the variance of the voxels' responses changed across the experiment.

## Recoding provides more efficient representation of multiple sequences than associative learning

To further explore why recoding might be an advantageous learning mechanism we contrasted the effective learning capacity between the two learning models. Specifically, we estimated how much would multiple to-be-learned sequences interfere with each other. For example, a single sequence can readily be learned by strengthening the item-position associations. However, such a coding scheme would struggle to effectively learn multiple overlapping sequences. For example, two sequences *ABCD* and *BADC* can be learned simultaneously by storing position-item associations (Fig 7A), but this would result in eight association weights of equal strength to represent four sequences (*ABCD*, *BADC* and *ABDC*, *BACD*; Fig 7B). Such a learning mechanism would suffer from catastrophic interference with multiple short sequences of overlapping items. Most naturally occurring sequences (words or sentences in a language, events in tasks like driving a car or preparing a dish) do not consist of items or events which only uniquely

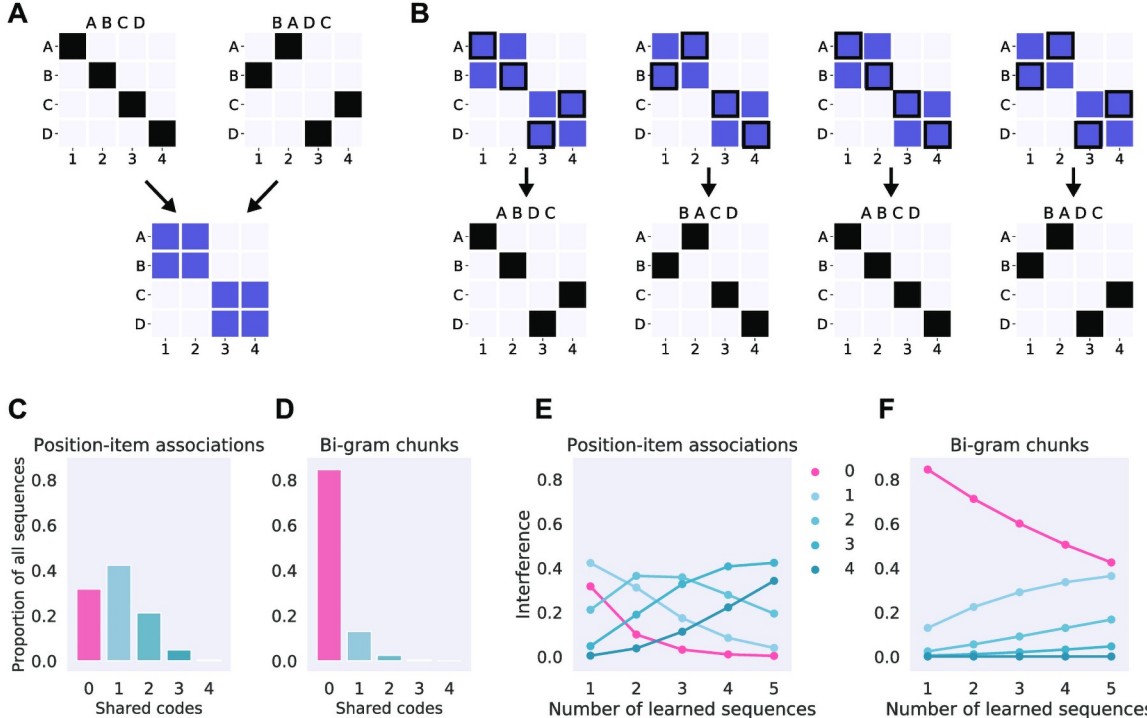

**Fig 7. Interference in sequence learning.** (A) Visual representation of two sequences as position-item associations (top) and the resulting frequency of associations (bottom) as defined by the associative sequence learning model. (B) Associative learning of two sequences on panel *A* would boost the representations of four individual sequences despite the statistical regularities being extracted from only two. See S4 Text for a worked example. (C) Histogram of the expected number of shared codes (item-position associations, x-axis) for a single 4-item sequence with all other possible 4-item sequences ($n = 256$, allowing repeats), measured as a proportion of sequences sharing the same number of codes (y-axis). (D) Histogram of the shared codes for a two-item (bi-gram) chunk representation. (E) Interference between sequence representations in the item-position model. X-axis displays how many sequences have been learned and lines on the plot display the proportion of other sequences affected by learning as a function of codes shared: the lines correspond to columns in panel *C*. The red line shows the proportion of sequences which have been unaffected by learning. (F) Interference between sequence representations in the chunk model.

occur in that sequence. Hence an efficient sequence learning mechanism has to be able to learn multiple overlapping sequences, which are re-orderings of the same items.

The interference resulting from strengthening of individual associations can be quantified for each learning model in terms of shared representations causing such interference. In the associative item-position model any individual sequence will have an expected similarity to all other possible sequences (Fig 7A). For example, *ABCD* shares two item-position associations with *DBCA*, and so forth. On the average, any individual 4-item sequence encoded with the item-position model shares two items with 21% of all the other possible sequences, while only 31% share no associations and about 5% share 3 out of 4 associations (Fig 7C). As more sequences are learned the interference between stored representations will inevitably increase since the number of possible associative codes in the item-position model is limited to the number of items × positions. Fig 7E shows that the effective capacity of learning different overlapping sequences with the associative item-position model is approximately five: at that point there are no sequences left which have been unaffected by learned sequences.

Contrastingly, a chunk recoding model that only uses two item chunks (bi-grams), has a markedly different expected similarity distribution (Fig 7D), resulting in significantly reduced interference between learned sequences (Fig 7F). Note that the bi-gram chunking model used here for illustrative purposes is the most limited chunking model possible: any flexible chunking model—such as the one estimated for our participants—will perform significantly better. A chunking model that is free to infer any number of chunks of any length can represent any number of multiple overlapping sequences without interference [39].

In sum, associative learning employs fewer codes but therefore necessarily loses in the representational power or 'coverage' over multiple overlapping sequences. Contrastingly, chunking allows emergence of more dissimilar codes which can be used to cover the space of all possible sequences with little interference.

## Discussion

In the current study we contrasted two classes of sequence learning models. First, we considered associative learning that proceeds by changing the signal-to-noise ratio of existing representations. Alternatively, repeated presentations might lead the initial representations to be recoded into more efficient representations such as chunks. Both mechanisms would result in more efficient codes and improve performance in the task: by either reducing uncertainty in the internal representations (associative learning) or reducing the necessary number of associations (chunk recoding). However, the two accounts make different predictions about changes the similarity between novel and repeating sequences.

### Learning induces recoding of sequence representations

We found that *novel* visual sequences were represented as position-item associations in a number of anatomically distinct regions in the dorsal visual processing stream. This is in line with previous research reporting that initial sequence representations are associative, binding individual events to a temporal order signal which specifies their position in a sequence [40, 41]. However, we found no evidence that *repeated* sequences were also represented positionally, as would be predicted by the associative learning model. Instead, we observed that learning proceeds by recoding the initial stimuli using a different set of codes. Specifically, the similarity between repeated and novel sequences followed predictions of the optimal chunking model in three cortical regions in the dorsal visual processing stream.

Such flexible recoding of stimuli in response to the changing statistics of the environment is a common and often necessary feature of probabilistic learning models (see [5] for a review).

However, most neural learning models assume that different populations represent different stages of learning: for example, a traditional hippocampal-cortical learning account assumes that the fast acquisition of initial associations is supported by the dynamics of the hippocampus proper while the cortical areas encode the consolidated representations [42]. Here we show that the same cortical region encodes both initial and learned representations.

Our findings are also consistent with behavioural data on memory for sequences where there is evidence for the use of positional coding when recalling novel sequences [43] while learned verbal sequences show little indication of positional coding [25].

## Recoding provides more efficient encoding of multiple overlapping sequences

Recoding initial sequence representations is also advantageous from an efficient coding perspective: we show that in our task an associative learner would be only able to effectively learn very few overlapping sequences, as it is limited by the space of possible associations. Contrastingly, recoding by chunking creates higher-dimensional codes, which can effectively store a limitless number of overlapping sequences [5, 10, 39]. Although higher-dimensional codes require more information to store than simpler low-dimensional associations, they are necessary to cover the vast space of possible overlapping sequences present in ecological working memory tasks such as reading, speaking, or navigating.

Importantly, sequence learning through recoding is likely only one of the multiple learning processes taking place in parallel. For example, accumulating evidence points to several streams of subcortical learning—mediated by the hippocampal formation and the basal ganglia—operating simultaneously [44–47] and the effects of both types of learning can be delineated for a single task in rodents [48]. Although in our task an efficient learner would benefit from recoding overlapping sequences, recoding might not offer significant advantages over associative learning in tasks with existing high-dimensional representations [49–51].

## Multiple and parallel systems for sequence learning

Our experimental task is significantly different from standard motor-sequence learning paradigms where learning proceeds through repetition of movements and consolidation can take several hours or days [24, 52, 53]. Here we used a working memory task with serial recall where individual sequences are typically learned in as few as 2–4 repeated presentations [54, 55] and learning proceeds even when no recall is attempted [56, 57]. Therefore, we think it is likely that in our task the change of representations happens at the level of working memory: the initial associative working memory representations are recoded with chunks.

Second, in our task visual presentation was followed by a brief delay and then a manual reproduction of the presented sequence. Therefore, we optimised the anatomical 'coverage' of the brain provided by the MRI scanner for the dorsal visuo-motor areas, where the anatomical locus of task representations should be expected: perceptual representations in the primary visual cortex [58], the delay-phase activity in the occipito-parietal and motor-parietal regions [59–62], and the manual response activity in the motor and pre-motor areas [20, 59]). Our MRI coverage did therefore not reach the deep subcortical regions such as the inferior and medio-temporal lobes and the basal ganglia, which have been shown to be crucial in both initial sequence encoding and learning (see S1 Fig). For example, a well-established model of neural learning [42, 47] proposes that the initial hippocampal-dependent associative memories are recoded as they undergo consolidation resulting in more efficient cortical representations [63, 64]. Specifically, while the initial encoding of individual events or items into novel sequences is facilitated by the hippocampal formation [65–67], learning modulates the parallel cortical

representations so that they become independent of the hippocampus over time: humans and animals with hippocampal lesions are unable to learn new sequences but are able to recall already learned ones [66, 68]. This suggests a dissociation between the hippocampal and cortical representations: both are created simultaneously but affected differently by learning.

Although we observed learning effects in the dorsal visual stream, we think it is very likely that similar effects could be observed in subcortical and frontal regions, given the fundamental role of these brain regions in structure learning. However, further research focussing on the direct comparison between the cortical and subcortical sequence representations is necessary to shed light on this relationship.

**Alternative learning models.** Associative learning and recoding, as two broad model classes, cover the whole space of statistical learning mechanisms for sequence representations. This is because all learning mechanisms can be labelled according to the simple distinction of whether the initial dimensionality of representations changes or not [69]. However, within each class we could construct different implementations, which would in turn lead to different predictions of similarity between sequences. For example, a chunk recoding model using only 4-item chunks would lead to significantly different predictions in our task compared to the optimal chunking model that we used. However, we designed our study so that a single chunking model would stand out in terms of prior probability since we wanted to maximise the chances of detecting the representational change. This meant that our stimuli were suboptimal for comparing individual chunking models: instead, longer and more individual learned sequences would be required [70–72].

Another common question raised with learning studies is to what extent could the results be explained by attentional differences between novel and learned stimuli: could participants allocate fewer resources to learned sequences once they have been identified as such? In our view such attentional effects are entirely expected in any statistical learning task, given that an optimal learner should allocate resources to stimuli proportional to the amount of information they provide about the environment [35]. Therefore, we would expect to see differential resource allocation in encoding novel and learned sequences, since learned sequences have higher prior probability and thus provide relatively little new information about the environment compared to novel ones. Since the probability of learned sequences is necessarily higher for any possible learning mechanism it follows that any learning model would predict different resource requirements. Univariate effects of resource allocation—decrease in the BOLD signal or reaction times, changes in perceived task difficulty—should be expected for any learning mechanism and do not allow dissociating between different computations underlying learning. Hence here we contrasted models which predict similar differences in of resource allocation but make dissociable predictions in terms of the dimensionality of neural responses. The associative learner would not change the representations (same dimensionality) while recoding would: this distinction is orthogonal to that of resource requirements.

## Conclusions

Our results suggest that humans follow an efficient sequence learning strategy and recode initial sequence representations into chunks. We found no evidence for the hypothesis that learning involves strengthening existing associations in our task. Furthermore, we show that associative learning without recoding is not theoretically capable of supporting long-term storage of multiple overlapping items. Although the initial associative representations of novel sequences may be sufficient to support immediate recall, multiple sequences can only be learned by developing more efficient higher order representations such as chunks.

## Methods

### Ethics statement

The study was approved by the Cambridge Local Research Ethics Committee (CPREC, Cambridge, UK; application PRE.2017.024). Every participant gave a formal written consent to take part of the study.

### Participants

In total, 25 right-handed volunteers (19–34 years old, 10 female) gave informed, written consent for participation in the study after its nature had been explained to them. Participants reported no history of psychiatric or neurological disorders and no current use of any psychoactive medications. Three participants were excluded from the study because of excessive inter-scan movements (see fMRI data acquisition and pre-processing).

### Task

On each trial, participants saw a sequence of items (oriented Gabor patches) displayed in the centre of the screen (Fig 2A). Each item was displayed on the screen for 2.4s (the whole four-item sequence 9.6s). Presentation of a sequence was followed by a delay of 4.8s during which only a fixation cue '+' was displayed on the screen. After the delay, participants either saw a response cue '*' in the centre of the screen indicating that they should manually recall the sequence exactly as they had just seen it, or a cue '–' indicating not to respond, and to wait for for the next sequence (rest phase; 10–18s). We used a four-button button-box where each button was mapped to a single item (see Stimuli below).

The recall cue appeared on 3/4 of the trials and the length of the recall period was limited to 7.2s. We omitted the recall phase for 1/4 of the trials to ensure a sufficient degree of de-correlation between the estimates of the BOLD signal for the delay and recall phases of the task. Each participant was presented with 72 trials (36 trials per scanning run) in addition to an initial practice session outside the scanner. In the practice session participants had to recall two individual sequences 12 times as they learned the mapping of items to button-box buttons. Participants were not informed that there were different types of trials.

**Stimuli.** All presented sequences were permutations of the same four items (see Sequence generation and similarity). The items were Gabor patches which only differed with respect to the orientation of the patch. Orientations of the patches were equally spaced (0, 45, 90, 135 degrees) to ensure all items were equally similar to each other. The Gabor patches subtended a 6˚ visual angle around the fixation point in order to elicit an approximately foveal retinotopic representation. Stimuli were back-projected onto a screen in the scanner which participants viewed via a tilted mirror.

We used sequences of four items to ensure that the entire sequence would fall within the participants' short-term memory capacity and could be accurately retained in STM. If we had used longer sequences where participants might make errors (e.g. 8 items) then the representation of any given sequence would necessarily vary from trial to trial, and no consistent pattern of neural activity could be detected. All participants learned which four items corresponded to which buttons during a practice session before scanning. These mappings were shuffled between participants (8 different mappings) and controlled for heuristics (e.g. avoid buttons mapping orientations in a clockwise manner).

**Sequence generation and similarity.** We permuted the same four items (oriented Gabor patches) into different individual sequences to resemble sequences in the natural world, which

are mostly permutations of a small set of items or events based on the desired outcome (e.g. driving the car, parsing a sentence, etc).

We chose the 14 individual four-item sequences used in the experiment (2 repeating, 12 unique) to be as dissimilar to each other as possible in order to avoid significant statistical regularities between individual sequences themselves and instead be able to introduce regularities only through repeating the individual sequences (see Chunk learning for details).

We constrained the possible set of individual sequences with two criteria:

1. *Dissimilarity between all individual sequences*: all sequences needed to be at least three edits apart in the Hamming distance space (see Similarity between sequence representations for details on the Hamming distance between two sequences). For example, given a repeating sequence {*A*, *B*, *C*, *D*} we wanted to avoid a unique sequence {*A*, *B*, *D*, *C*} as these share the two first items and hence the latter would only be a partially *unique* sequence. This would allow in chunk learning to encode both sequences with a common multi-item chunk *AB*.

2. *N-gram dissimilarity between two repeating sequences*: the two repeating sequences shared no items at common positions and no common n-grams, where $n > 1$ (see Chunk learning for n-gram definition and details). This ensured that the representations of repeating sequences would not interfere with each other and hence both could be learned to similar level of familiarity. Secondly, this ensured that for the chunking model the repetitions of these two sequences were optimally encoded with two four-grams since they shared no common bi-grams of tri-grams.

Given these constraints, we wanted to find a set of sequences which maximised two statistical power measures:

1. *Between-sequence similarity score entropy*: this was measured as the entropy of the lower triangle of the between-sequence similarity matrix. The pairwise similarity matrix between 14 sequences has $14^2 = 196$ cells, but since it is diagonally identical only 84 cells can be used as a predictor of similarity for experimental data. Note that the maximum entropy set of scores would have an equal number of possible distances but since that is theoretically impossible, we chose the closest distribution given the restrictions above.

2. *Between-model dissimilarity*: defined as the correlation between pairwise similarity matrices of different sequence representation models (see Similarity between sequence representations). We sought to maximise the dissimilarity between model predictions, that is, decrease the correlation between similarity matrices.

The two measures described above, together with the constraints, were used as a cost function for a grid search over a space of all possible subsets of fourteen sequences ($k = 14$) out of possible total number of four-item sequences ($n = 4!$). Since the Binomial coefficient of possible choices of sequences is ca $2 \times 10^6$ we used a Monte Carlo approach of randomly sampling $10^4$ sets of sequences to get a distribution for cost function parameters. This processes resulted in a set of individual sequences which were used in the study (see S6 Text).

**Structure of the trials.**   To test our hypotheses we split the 14 individual sequences in to two classes: two of these were repeatedly presented throughout the experiment (*repeating* sequences, 2/3 of the trials) while the remaining 12 were previously unseen and were only presented once (*unique* sequences, 1/3 of the trials). The two individual repeating sequences were chosen randomly for each participant.

The two *repeating* sequences were also used for training before the scanning experiment (each presented 12 times). This was done to ensure that the two repeating sequences would be

12 times more likely already at the start of the experiment and stay so throughout scanning (see Optimal chunking model for details).

To keep the relative probability of repeating and unique sequences fixed throughout the experiment we divided the trials into sub-blocks of three trials so that each contained a single unique sequence and two repeating sequences in random order (Fig 2B). This ensured that after every three trials the participant exposure to repeated and unique sequences was the same (2/3 repeated, 1/3 unique sequences).

For MRI scanning we repeated this experimental block twice for every participant so that in a 36-trial scanning session participants recalled each unique sequence once and repeating sequences 12 times each (Fig 2B). Over two scanning sessions this resulted in 48 trials with repeating sequences and 24 trials with unique sequences.

## Sequence representation models

Sequences are associative codes: they are formed either by associating successive items to each other (item-item associations) or by associating items to some external signal specifying the temporal context for sequences (item-position associations).

In the case of item-position associations sequences are formed by associating items to some external signal specifying the temporal context. This context signal can be a gradually changing temporal signal [73–75], a discrete value specifying the item's position in a sequence [76], or a combination of multiple context signals [77, 78]. Common to all of these models is the underlying association of item representations to the positional signal, forming item-position associations (Fig 1A). Alternatively, for item-item associations the weights of the associations are usually expressed in terms of transitional probabilities [1] forming a 'chain' of associations [79]. Past research has provided evidence for both: sequences are represented as item-position associations in rodent, primate, and humans brains [80–82] and also as item-item associations [83] depending on task type and anatomical area (see [1] for a review).

For our sequence processing task (Fig 2) we model the participants' internal sequence representations $\mu$ given the presented sequence $y$ as Bayesian inference (Eq 2), where the posterior distribution $p(\mu|y)$ represents a participant's estimate of the presented stimulus, and their response can be thought of as a sample from the posterior distribution:

$$\overbrace{p(\mu|y)}^{posterior} \propto \overbrace{p(y|\mu)}^{likelihood} \cdot \overbrace{p(\mu)}^{prior} \tag{2}$$

Associations between discrete variables—such as items, or items and positions—can be formalised as a multinomial joint probability distribution. The multinomial representation can in turn be visualised as a matrix where each cell describes the probability of a particular item at a particular position (Fig 1).

Formally, every item $x$ in the sequence $z$ is represented by a multinomial variable which can take $K$ states parametrised by a vector $\boldsymbol{\mu} = (\mu_1, \ldots, \mu_K)$ which denotes the probability of item $x$ occurring at any of $k$ positions:

$$p(\mathbf{x}|\boldsymbol{\mu}) = \prod_{k=1}^{K} \mu_k^{x_k}, \tag{3}$$

and the whole $N$-item sequence $\mathbf{z} = (x_1, \ldots, x_N)^T$ is given by:

$$p(\mathbf{z}|\boldsymbol{\mu}) = \prod_{n=1}^{N} \prod_{k=1}^{K} \mu_k^{x_{nk}}, \tag{4}$$

where the $\boldsymbol{\mu}$ represents the probability of particular item-position associations and hence must satisfy $0 \leq \mu_k \leq 1$ and $\Sigma_k \, \mu_k = 1$. Exactly the same formalism applies to item-item associations: we simply replace the set of $K$ position variables with another identical set of $N$ items.

**Similarity between sequence representations.**

*Item-position associations*

When sequences are represented as item-position associations they can be described in terms of their similarity to each other: how similar one sequence is to another reflects whether common items appear at the same positions. Formally, this is measured by the Hamming distance between two sequences:

$$D_H(\mathbf{y}_j, \mathbf{y}_l) \quad = \sum_{i=1}^{k} |x_j^i - x_l^i| \tag{5}$$

$$x_j^i = x_l^i \Rightarrow 0 \tag{6}$$

$$x_j^i \neq x_l^i \Rightarrow 1 \tag{7}$$

where $x_j^i$ and $x_l^i$ are the $i$-th items from sequences $\mathbf{y}_j$ and $\mathbf{y}_l$ of equal length $k$. Consider two sequences *ABCD* and *CBAD*: they both share two item-position associations (*B* at the second and *D* at the fourth position) hence the Hamming distance between them is 2 (out of possible 4).

We use the between-sequence similarity as defined by the Hamming distance as a prediction about the similarity between fMRI activation patterns: if sequences are coded as item-position associations then the similarity of their corresponding neural activity patterns, all else being equal, should follow the Hamming distance. This allows us to test whether a particular set of voxels encodes information about sequences using an item-position model. Representational similarity analysis of fMRI activity patterns below provides the details of the implementation.

*Item-item associations*

Here we use n-grams as associations between multiple consecutive items to define sequences as pairwise item-item associations: a four-item sequence *ABCD* can be unambiguously represented by three bi-grams *AB*, *BC*, *CD* so that every bi-gram represents associations between successive items. The bi-gram representation of item-item associations can be used to derive a hypothesis about the similarity between sequences: the between-sequence similarity is proportional to how many common item pairs they share. For example, the sequences *FBI* and *BIN* both could be encoded using a bi-gram where *B* is followed by *I* (but share no items at common positions and are hence dissimilar in terms of item-position associations). This allows us to define a pairwise sequence similarity measure which counts how many bi-grams are retained between two sequences:

$$S_C(S_i, S_j) = \text{card}(C_i \cap C_j) \tag{8}$$

where $C_i$ and $C_j$ are the sets of n-grams required to encode sequences $S_i$ and $S_j$ so that *card* $(C_i \cap C_j)$ denotes the cardinality of the union of two sets of n-grams (i.e. the number of elements in the resulting set). All possible constituent n-grams of both sequences can be extracted iteratively starting from the beginning of sequence and choosing $n$ consecutive items as an n-gram. For bi-grams this gives:

$$C_i = \{i = 1, \ldots, k - 1 \;\; : (x_i, x_{i+1})\}$$

where $C_i$ is a set of all possible adjacent n-grams from sequence $S_i$ of length $k$ so that every bi-gram is a pair (tuple) of consecutive sequence items $(x_i, x_{i+1})$. Similarly for a set of n-grams $C$ from any sequence of length $k$:

$$C = \{i = 1, \ldots, k - (n - 1) \ : (x_i, \ldots, x_{i+(n-1)})\},$$

where $n$ is the length of n-gram. Effectively, the n-gram similarity counts the common members between two n-gram sets. Given sequence length $k$ this similarity can accommodate n-grams of all sizes $n$ (as long as $n \leq k$). To make the measure comparable for different values of $n$ we need to make the value proportional to the total number of possible n-grams in the sequence and convert it into a distance measure by subtracting it from 1:

$$D_C = 1 - \gamma \ \mathrm{card}(C_i \cap C_j) \tag{9}$$

where $\gamma$ is a normalising constant:

$$\gamma = \frac{1}{k - (n - 1)}.$$

Effectively, the n-gram distance $D_C$ counts the common members between two n-gram sets. We then used the bi-gram distance measure to derive sequence representation predictions for item-item association models.

The prediction made by the n-gram distance $D_C$ is fundamentally different from the prediction made by the Hamming distance $D_H$ (Eq 7): the n-gram distance assumes that sequences are encoded as item-item associations whilst the Hamming distance assumes sequences are encoded as item-position associations.

To understand why the item-position and item-item models make inversely correlated predictions, consider again the example given above: two sequences of same items *FBI* and *BIF* are similar from a bi-gram perspective since both could be encoded using a bi-gram where $B$ is followed by $I$ (but share no items at common positions and are hence dissimilar in terms of item-position associations). Conversely, two sequences *FBI* and *FIB* share no item pairs (bi-grams) and are hence dissimilar form a bi-gram perspective but have both $F$ at the first position and hence somewhat similar in terms of the item-position model (Hamming distance).

*Item mixture*

We also defined an additional control model which tested for a null-hypothesis that instead of sequence representations neural activity could be better explained by the superposition of patterns for constituent individual items in the sequence, called the *item mixture model* (e.g. see [22]).

This model is not a sequence representation model but rather an alternative hypothesis of what is being captured by the fMRI data. This model posits that instead of sequence representations fMRI patterns reflect item representations overlaid on top of each other like a palimpsest so that the most recent item is most prominent. For example, a sequence *ABCD* could be represented as a mixture: 70% the last item ($D$), 20% the item before ($C$), and so forth. In this case the mixing coefficient increases along the sequence. Alternatively, the items in the beginning might contribute more and we would like to use a decreasing mixing coefficient. If all items were weighted equally the overall representations would be identical as each sequence is made up of the same four items. Here we only considered linearly changing coefficients: we did not consider non-linear or random weights.

Formally, we model an item mixture representation $M$ of a sequence as a weighted sum of the individual item representations:

$$M = \mathbf{I}\beta \qquad (10)$$

where $\mathbf{I}$ is the four-dimensional representation of individual items in the sequence and $\beta$ is the vector of mixing coefficients so that $\beta_n$ is the mixing coefficient of the $n$-th item in $\mathbf{I}$ so that

$$0 < \beta_n \leq 1, \text{ and } \sum_{m=1}^{N} \beta_n = 1.$$

where $N$ is the length of the sequence. The rate of change of $\beta$ (to give a particular $\beta_n$ a value) was calculated as

$$\beta_n = \alpha\beta_0(1-\theta)^n,$$

where $\theta$ is the rate of change and $\alpha$ normalising constant. In this study we chose the value of $\theta$ so that $\beta = \{0, 1/6, 1/3, 1/2\}$ represents a recency-weighted slope over individual sequence items. The reason we only tested for the 'recency mixture' is that the distances between mixtures only depend on the absolute value of the slope of the mixture coefficients over the items. In other words, an RDM derived with a recency-based item mixture predicts the same similarity between voxel patterns as an RDM derived with a primacy based mixture given the absolute value of the gradient slope remains the same. Here we chose a middle point between two extreme slope values: all the mixtures become the same when the slope is horizontally flat and only a single item contributes when the slope is vertical. See S2 Text for more details and a worked example.

Distances between two item mixture representations $M_i$ and $M_j$ (Eq 10) of sequences $S_i$ and $S_j$ were calculated as correlation distances:

$$D_I(S_i, S_j) = cdist(M_i, M_j). \qquad (11)$$

## Associative learning

The optimal way of encoding how many times particular discrete associations have occurred is given by the Dirichlet-Multinomial model. In short, past occurrences of items at certain positions are transformed into probabilities, which reflect the frequency of associations. Hence associative learning can be thought of as changing the weights of associations—$\boldsymbol{\mu}$ parameter in the multinomial model above—so that they reflect the statistics of the environment. This is achieved by deriving $p(\boldsymbol{\mu})$ from the Dirichlet distribution:

$$\boldsymbol{\mu} \sim \text{Dir}(\boldsymbol{\alpha}) \qquad (12)$$

where $\alpha = (\alpha_1, \ldots, \alpha_K)^T$ denotes the effective number of observations for individual associations. The optimal internal representation of associations for a sequence $\boldsymbol{y}$ is therefore given by:

$$p(\boldsymbol{\mu}|\boldsymbol{y}, \boldsymbol{\alpha}) \propto p(\boldsymbol{y}|\boldsymbol{\mu})p(\boldsymbol{\mu}|\boldsymbol{\alpha}). \qquad (13)$$

We could also use $\boldsymbol{\mu}$ to introduce additional biases into the model (e.g. recency or primacy effects) but since our task has short sequences and clearly distinctive individual items such additional biases are not significant (see Behaviour in Results). In formal terms this means specifying the conjugate prior for the parameter $\mu$ of the multinomial prior

distribution (Eq 4), which is given by the Dirichlet distribution:

$$p(\mu|\alpha) = \phi \prod_{k=1}^{K} \mu_k^{a_k - 1},$$

$$(14)$$

where $0 \leq \mu_k \leq 1$ and $\Sigma \mu_k = 1$ and $\phi$ is the normalisation constant. The parameters $\alpha_k$ of the prior distribution can be interpreted as an effective number of observations $x_k = 1$, or in other words, the counts of the value of the sequence position $x$ previously. Effectively, the conjugate prior tracks the item-position occurrence history. Since this model reflects the expected value of item-position associations it is also an optimal model of sequence representation assuming that associations are independent of each other.

Using position-item associations as defined above to encode a set of individual sequences **S** = (*BACD*, *CABD*, *ABCD*) will result in a following value for $\mu$ reflecting the position-item counts:

$$\mu = \begin{pmatrix} 1/3 & 2/3 & 0 & 0 \\ 1/3 & 1/3 & 1/3 & 0 \\ 1/3 & 0 & 2/3 & 0 \\ 0 & 0 & 0 & 1 \end{pmatrix}$$

Here matrix rows and columns reflect the items and position variables. Changing or adding new sequences to the set will only change the probabilities or association weights but not change individual items bound by associations. This matrix is visualised for three item-position associations in Fig 1C.

## Chunk learning

A chunking model $\theta_i$ is defined by two parameters and their probability distributions $p(\mathbf{x}, \mathbf{z}|\theta_i)$, where $\mathbf{x}$ is a set of individual chunks and $\mathbf{z}$ a set of mappings defining how chunks are arranged together to encode observed sequences.

*Set of chunks*

We represent chunks formally as n-grams (used from hereon synonymously with the term 'chunk') that can take any length up the maximum possible length of a sequence to be encoded. For illustrative purposes we denote the individual items in our sequences here with letters: a four-item sequence of Gabors can be written as *ABCD* and in turn be represented by a tri-gram *ABC* and a uni-gram *D*. For 4-item sequences the set of all possible n-grams has the number of *P* members as the sum of partial permutations of $n = 4$ items taken $k = \{1, 2, 3, 4\}$ at a time:

$$P_N = \sum_{k=1}^{n} \frac{n!}{(n-k)!} = 64$$

A set of chunks $\mathbf{x}$ comprises $J$ n-grams where each constituent n-gram $c$ appears only once: $\mathbf{x} = \{c_1, \ldots, c_J\}$, and $1 < J < P_N$; for example $\mathbf{x} = \{AB, BA, A, B, CDA, ACDB\}$. Each individual n-gram has a probability inversely proportional to its combinatorial complexity. Specifically, the probability of a particular n-gram $c_j$ is proportional to the reciprocal of the number of partial permutations of $n = 4$ items taken $k$ at a time:

$$p(c_j) = \alpha \frac{1}{\frac{n!}{(n-k)!}},$$

$$(15)$$

where $k$ is the length of the n-gram and $\alpha$ is the normalising constant. For example, there are 4 possible uni-grams for a 4-item sequence, but 12 bi-grams, 24 tri-grams, etc. Hence the probability of a uni-gram is 3 times greater than a bi-gram and so forth. This also captures the intuition that longer and more complex chunks should be less likely than simple chunks. We also assume that chunks are independent each other and hence the probability of a set of n-grams $\mathbf{x}$ defined by the chunking model is the product of its constituent chunk probabilities:

$$p(\mathbf{x}|\theta_i) = \prod_{j=1}^{J} p(c_j). \tag{16}$$

*Mappings between chunks*

The second parameter of the chunking model describes how individual n-grams are combined together to encode the observed sequences. For example, given a single sequence *ABCD* and a set of n-grams $\mathbf{x}$ = {*AB*, *BC*, *CD*, *A*, *B*, *C*, *D*} we can encode the data as (*AB*, *CD*) or (*A*, *BC*, *D*) as both mappings are capable of representing the observed data without error.

For any 4-item sequence there are eight possible ways n-grams can be linked together to reproduce the observed sequence. These mappings can be described as a set of eight tuples $\mathbf{Z}$ = $\{\mathbf{g}_1, \ldots, \mathbf{g}_8\}$, where each tuple defines $F \leq 4$ links $\mathbf{g}_i = ((l_1), \ldots, (l_F))$ that exhaustively define all possible n-gram parses of a 4-item sequence:

$$\mathbf{Z} = \begin{aligned} &\{((1), (2, 3), (4)), \\ &((1), (2, 3, 4)), \\ &((1, 2, 3), (4)), \\ &((1, 2), (3), (4)), \\ &((1), (2), (3), (4)), \\ &((1), (2), (3, 4)), \\ &((1, 2, 3, 4)), \\ &((1, 2), (3, 4))\} \end{aligned}$$

For example, given a sequence *ABCD*, the first tuple in the set $\mathbf{g}_1$ = ((1,), (2,3), (4,)) corresponds to linking three individual n-grams as ((*A*), (*B*, *C*), (*D*)). The mappings $\mathbf{g}_i$ in $\mathbf{Z}$ differ in terms of how many links are required to encode the sequence: for example, the first mapping comprises three links, the second two, and the fifth four. The probability of each mapping is a product of it's individual link probabilities:

$$p(\mathbf{g}_i) = \prod_{j=1}^{F} p(l_j) = \eta^F, \tag{17}$$

where $F$ is the number of links in the mapping $\mathbf{g}_i$ and $\eta$ is a probability of each link which we assume to be constant (the reciprocal of the number of possible links). Such inverse relationship between the probability of a mapping and its length captures the intuition that complex mappings between multiple n-grams should be less likely than simple ones. Note that for longer sequences a different relationship might be defined as the ability to combine chunks is limited by human short term memory capacity which sets an effective limit to the length of sequence that can be encoded [84–86].

For a particular model $\theta_i$ the mapping parameter $\mathbf{z}$ defines how n-grams are combined together to generate observed sequences. For example, consider two models and a set of sequences $\mathbf{S}$ = {*ABCD*, *ABCD*, *CDAB*}. Both models use the same set of n-grams $\mathbf{x}$ = {*AB*, *CD*,

$A$, $B$, $C$, $D$}, but encode the observed sequences differently:

$$\mathbf{z}_1 = \{((A,B),(C,D)),((A,B),(C,D)),((C,D),(A,B))\}$$
$$\mathbf{z}_2 = \{((A),(B),(C),(D)),((A),(B),(C),(D)),((C),(D),(A),(B))\}$$

Although these two models are equally likely in terms of the chunks they employ, their mappings have different probabilities. The probability of mappings defined by a particular model is equal to the product of mappings for individual sequences:

$$p(\mathbf{z}|\theta_i) = \prod_{i=1}^{M} p(\mathbf{g}_i), \tag{18}$$

where $M$ is the number of mappings (sequences encoded).

For any model $\theta_i$ the probability of both parameters—n-grams and mappings—are assumed to be independent of each other and therefore the probability of any particular model is the product their parameter probabilities:

$$p(\mathbf{x}, \mathbf{z}|\theta_i) = \prod_{j=1}^{J} p(c_j) \prod_{i=1}^{M} p(\mathbf{g}_i), \tag{19}$$

where $J$ and $M$ are the number of n-grams and mappings. Therefore every model and its two parameters propose an encoding based on some chunks (e.g. examples above), which can subsequently compared to the observed data.

**Optimal chunking model.** The optimal model can be estimated by either randomly sampling the parameter distributions or using a systematic approach. Here we found the optimal model by only considering models that result in parsing the set of sequences into chunks and creating a ranking based on model evidence: see S3 Text for details. Furthermore, we designed the experiment so that the only regularities between the sequences that could be encoded with common chunks arose from repeating the same two sequences: this ensured that we effectively knew the optimal model beforehand.

To recall, our task included 14 individual sequences made maximally dissimilar to each other with two of them repeated on 2/3 of the trials. We first presented the two repeating sequences 12 times each during the practice session immediately before the experiment. Since those two sequences shared no common bi-grams or tri-grams (see Sequence generation and similarity) the only efficient chunk encoding for the set of repetitions of those two sequences comprised just two four-item chunks (Fig 8A). Using 12 repetitions in the training session was enough to make the four-gram representation the most likely one (Fig 8B). Therefore at the start of the experiment (before the first trial $t = 1$) the optimal chunking model had a set chunks defined as:

$$\mathbf{x}_{t=0}^{MAP} = \{CADB, DCBA\},$$

and a set of mappings $\mathbf{z}_{t=0}^{MAP}$ where the set of 24 sequences (made up of just two individual sequences) were encoded with the same one-on-one mappings.

We proceeded to estimate the optimal chunking model at every trial $t$ as the set of sequences was updated with newly observed stimulus. For this purpose we kept the statistical structure of the sequences fixed across the experiment: otherwise an optimal model on trial one would be different to the one on the last trial. Therefore we organised the order of trials so that on the average there was a single unique sequence and two repeating sequences in three consecutive trials. This ensured that after every three trials the participant exposure to repeated and unique sequences was roughly the same.

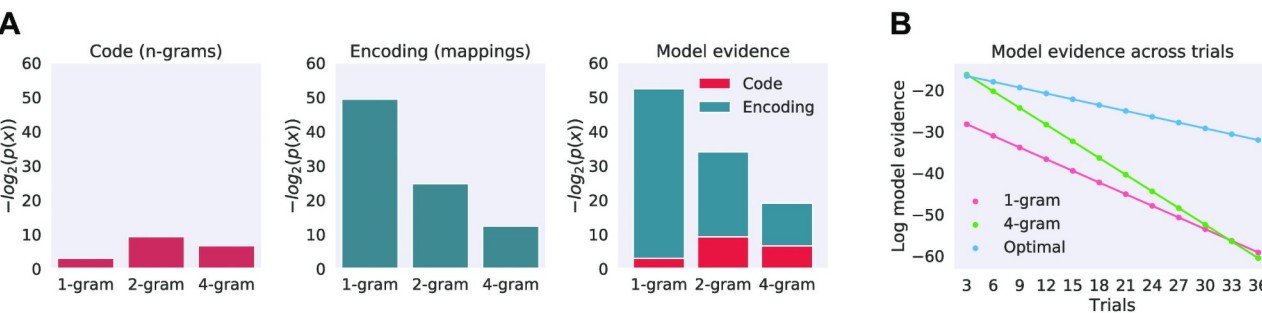

**Fig 8. Optimal chunking model.** (A) Evidence for three alternative chunking models and their components at the beginning of the scanning experiment, when participants had seen the two repeating sequences 12 times each during the practice session. The three models use only single type n-grams: the 1-gram model encodes sequences using four single-item n-grams, 2-gram model with two bi-grams, and the 4-gram model with a single four-gram. The left panel shows the probability of the set of n-grams (code) each model specifies in terms of their negative log values. The centre panel shows the probability of their mappings (encoding) and the right panel the combination of the two into model evidence. The blue and red parts of the model evidence bar represent model code (n-grams) and encoding (mappings) probabilities in terms of their negative logs and the total length of the bar displays the model evidence as their sum. This allows intuitive visualisation of the code-encoding trade-off calculated by the Bayesian model comparison. The 4-gram model is the optimal model at the start of the experiment. (B) Model evidence across trials. X-axis shows the trial number and y-axis shows the log of model evidence. The optimal model is inferred at every trial; the 1-gram model encodes sequences only with four uni-grams, and the 4-gram model only uses four-grams. Note that at the beginning of the experiment the 4-gram model is equivalent to the optimal model: however, as new sequences are presented the optimal model encodes new data with shorter chunks (uni-grams) while the 4-gram model encodes new unique sequences with four-grams. Note that as new data is observed the evidence for any particular model decreases as the set of data becomes larger and the space of possible models increases exponentially. Also note that the log scale transforms the change of evidence over trials into linear form.

Since the unique sequences shared no significant statistical regularities with each other or with the repeating sequences they could not have been efficiently encoded with common n-grams ($n > 1$). Therefore, at trial $t = 1$ the optimal model to encode the previously seen repeating sequences and the new unique one included the previously inferred two four-grams and additionally four single item uni-grams:

$$\mathbf{x}_{t=1}^{MAP} = \{CADB, DCBA, A, B, C, D\}. \tag{20}$$

Since we kept the statistical structure of the sequences fixed across the experiment this ensured that the optimal model would remain the same for every trial across the experiment: on a trial when a repeating sequence was presented it was encoded with a single four-gram chunk, and when a unique sequence was presented it was encoded with four uni-grams, as shown on Fig 8B.

## Representational similarity analysis (RSA)

**Representational similarity analysis of fMRI activity patterns.** First, we created a representational dissimilarity matrix (RDM) $\mathbf{S}$ for the stimuli by calculating the pairwise distances $s_{ij}$ between sequences $\{N_1, \ldots, N_M\}$:

$$\mathbf{S} = \begin{bmatrix} s_{1,1} & \cdots & s_{1,M} \\ \vdots & \ddots & \vdots \\ s_{M,1} & \cdots & s_{M,M} \end{bmatrix},$$

$$s_{ij} = D(N_i, N_j)$$

where $s_{ij}$ is the cell in the RDM **S** in row $i$ and column $j$, and $N_i$ and $N_j$ are individual sequences. $D(N_i, N_j)$ is the distance measure corresponding to any of the sequence representation models tested in this study:

1. The item-position model: Hamming distance (Eq 7)

2. The item-item model: bi-gram distance (Eq 9)

3. The item mixture model: the item mixture distance (Eq 11)

4. The optimal chunking model: n-gram distance (Eq 9) between the individual chunks

Next, we measured the pairwise distances between the voxel activity patterns:

$$\mathbf{A} = \begin{bmatrix} a_{1,1} & \cdots & a_{1,M} \\ \vdots & \ddots & \vdots \\ a_{M,1} & \cdots & a_{M,M} \end{bmatrix}, \tag{21}$$

$$a_{ij} = cdist(P_i, P_j) = 1 - \frac{(P_i - \mu_{P_i}) \cdot (P_j - \mu_{P_j})}{||(P_i - \mu_{P_i})||_2 ||(P_j - \mu_{P_j})||_2} \tag{22}$$

where $a_{ij}$ is the cell in the RDM **A** in row $i$ and column $j$, and $P_i$ and $P_j$ are voxel activity patterns corresponding to sequences $N_i$ and $N_j$. As shown by Eq 22, the pairwise voxel pattern dissimilarity is calculated as a correlation distance.

We then computed the Spearman's rank-order correlation between the stimulus and voxel pattern RDMs for every task phase $p$ and ROI $r$:

$$r_{r,p} = \rho(\mathbf{rS}_{r,p}, \mathbf{rA}_{r,p}) = \frac{\mathbb{E}[(\mathbf{rS}_{r,p} - \mu_{\mathbf{rS}_{r,p}})(\mathbf{rA}_{r,p} - \mu_{\mathbf{rA}_{r,p}})]}{\sigma_{\mathbf{rS}_{r,p}} \sigma_{\mathbf{rA}_{r,p}}} \tag{23}$$

where $\rho$ is the Pearson correlation coefficient applied to the ranks **rS** and **rA** of data **S** and **A**.

Finally, we tested whether the Spearman correlation coefficients $r$ were significantly positive across participants (see Significance testing below). The steps of the analysis are outlined on Fig 3.

**Noise ceiling estimation.** Measurement noise in an fMRI experiment includes the physiological and neural noise in voxel activation patterns, fMRI measurement noise, and individual differences between subjects—even a perfect model would not result in a correlation of 1 with the voxel RDMs from each subject. Therefore an estimate of the noise ceiling is necessary to indicate how much variance in brain data—given the noise level—is expected to be explained by an ideal 'true' model.

We calculated the upper bound of the noise ceiling by finding the average correlation of each individual single-subject voxel RDM (Eqs 21 and 22) with the group mean, where the group mean serves as a surrogate for the perfect model. Because the individual distance structure is also averaged into this group mean, this value slightly overestimates the true ceiling. As a lower bound, each individual RDM was also correlated with the group mean in which this individual was removed.

We also tested whether a model's noise ceiling was significantly greater than zero. We first Fisher transformed individual Spearman's rank-order correlation values and then performed a one-sided $t$-test for the mean being greater than zero. The 5% significance threshold for the $t$-

value was corrected for multiple comparisons (number of regions). For more information about the noise ceiling see [36].

In sum, we only considered a model evidence to be significant if it satisfied three criteria: (1) the model correlation with data was significantly greater across participants than the lower bound of the noise ceiling, (2) the lower bound of the noise ceiling was significantly greater than zero across participants, and (3) the average correlation for the item-mixture model (null-hypothesis) did not reach the noise ceiling.

### Associative learning predictions for RSA

Associative learning makes two predictions: learning doesn't change individual item representations and learning reduces noise in sequence representations. These hypotheses can be tested by measuring the similarity between neural activation patterns elicited by novel and repeating sequences.

Noise in sequence representations can be estimated by assuming that the voxel pattern similarity $\mathbf{A}$ (Eq 21) is a function of the 'true' representational similarity between sequences $\mathbf{S}$ plus some measurement and/or cognitive noise $v$: $\mathbf{A} = \mathbf{S} + v$. Here the noise $v$ is the difference between predicted and measured similarity. Note that this is only a valid noise estimate when the predicted and measured similarity are significantly positively correlated (i.e. there is 'signal' in the channel).

If learning reduces noise in sequence representations then the noise in activity patterns generated by novel sequences $v_N$ should be greater than for repeating sequences $v_L$. To test this we measured whether the activity patterns of repeating sequences were similar to novel sequences as predicted by the Hamming distance. The analysis followed exactly the same RSA steps as above, except instead of carrying it out within novel sequences we do this between novel and repeating sequences. First, we computed the Hamming distances between individual repeating and novel sequences $\mathbf{S}_{U,R}$, next the corresponding voxel pattern similarities $\mathbf{A}_{U,R}$ and finally computed the Spearman's rank-order correlation between the stimulus and voxel pattern RDMs exactly as above (Eq 23). If this measured correlation is significantly greater than the one within novel sequences ($r_{U,R} > r_U$) across participants, it follows that the noise level in repeating representations is lower than in novel representations. This analysis was carried out for all task phases and in all ROIs and the outcome could fall into one of three categories:

1. No significant correlation: the probability of $r_{U,R}$ is less than the significance threshold ($p < 10^{-4}$; see Significance testing below). This means that repeating sequences are not represented as item-position associations in this ROI and hence the test for noise levels is meaningless.

2. Significant correlation, but consistently smaller across participants than the within-novel sequences measure: $r_{U,R} < r_U$. repeating sequence representations are noisier than novel sequence representations.

3. Significant correlation, but consistently greater across participants than the within-novel sequences measure: $r_{U,R} > r_U$. repeating sequence representations are less noisy than novel sequence representations.

To confirm that our assumptions regarding the effects of noise on sequence representation were correct we estimated the fMRI measurement noise for the participants in our task and tested to what degree the noise should be reduced (or signal-to-noise ratio increased) in the fMRI patterns for the changes to be detectable with the representational similarity analysis. The details of this simulation can be found in S5 Text.

### Recoding predictions for RSA

The recoding model predicts that the repetition of individual sequences should recode individual associations of those sequences. We assumed that participants were ideal learners and inferred an optimal chunking model based on the statistics across previously seen sequences (Eq 20). See the Chunk learning and Optimal chunking model sections for estimation details. Importantly, we designed the presentation of repeating and novel sequences so that the optimal model would remain the same for every trial across the experiment: every repeating sequence was encoded with a single four-gram chunk, and every novel sequence with four uni-grams (Fig 4, bottom row). For every participant we then estimated an RDM predicting the distances across novel and repeating sequences using the n-gram distance method described above (Eq 9). First, we computed the n-gram distances between individual repeating and novel sequences $\mathbf{S}_{U,R}$, next the corresponding voxel pattern similarities $\mathbf{A}_{U,R}$ and finally computed the Spearman's rank-order correlation between the stimulus and voxel pattern RDMs exactly as above (Eq 23).

### Model-free fMRI analyses of learning effects

All analyses were carried out with pre-processed data as detailed in the Functional data pre-processing section, expect for the univariate analyses, where we used regression coefficient values (beta values) the group level analysis.

**Univariate analysis of novel vs. learned sequences.**   We fitted a GLM where every trial was labelled either novel or learned resulting in two regressors (beta values), plus additional nuisance regressors as described in the Functional data pre-processing section. The average difference between participants' beta values for every voxel was then tested with 2-sided $t$-test and controlled for multiple comparisons using the family-wise error rate as implemented in the SPM-12 package.

**Changes in pattern distances across the experiment.**

*Between two repeating sequences*

We computed voxel pattern distance between each of the $N$th repetition of the two repeating sequences and estimated a slope across repetitions (least squares linear regression) to see whether there was a significant change in distance across trials. Fig 9 below displays this for a

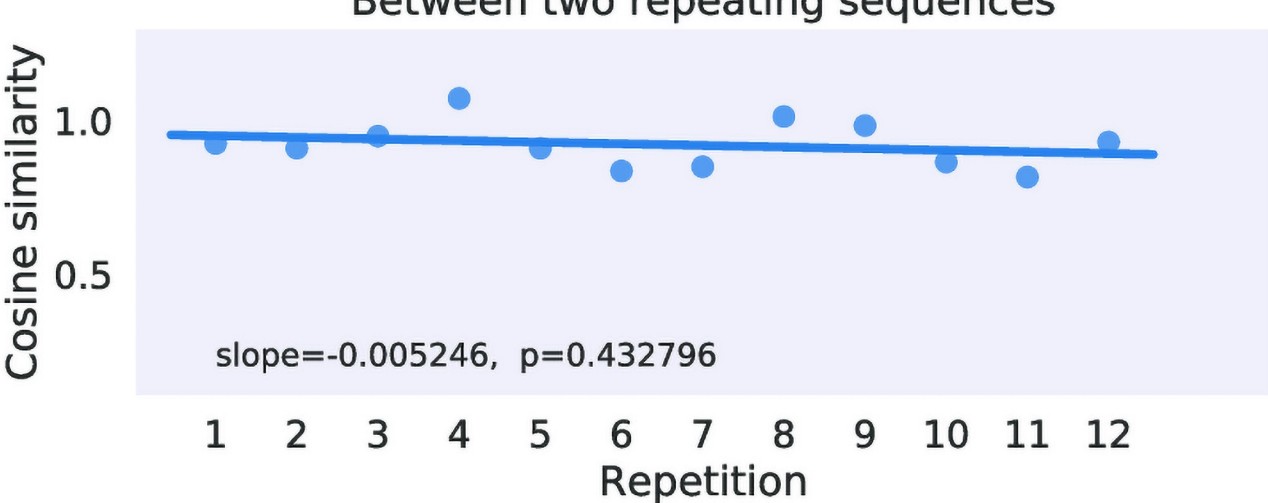

**Fig 9. Change in pattern distance across trials for a single participant and region.** *Y*-axis displays the distance value, while *x*-axis the trial number.

single participant and region: $y$-axis displays the distance value, while $x$-axis the trial number. For example, the data point at $x = 1$ represents the distance between two individual repeated sequences $R_1$ and $R_2$ at their first presentation, and all 12 data points are calculated as:

$$d_{x=1} = distance(R_1^1, R_2^1),$$

$$\ldots,$$

$$d_{x=12} = distance(R_1^{12}, R_2^{12})$$

where superscript denotes the repetition number and subscript the identity of the sequences. In all distance analyses we used the cosine distance between two patterns $u$ and $v$ defined as:

$$distance(u, v) = 1 - \frac{u \cdot v}{u_2 v_2}$$

We then tested whether the participants' slope values were significantly different from zero for all ROIs. The significance threshold was $\alpha = 0.05/(\text{Number of ROIs})$.

*Within repeating sequences*

We measured whether voxel pattern distances within the individual two repeating sequences changed significantly across the experiment. This test measured whether there was a change in distances between consecutive presentations of the same individual repeated sequence:

$$d_n = distance(R^n, R^{n-1}).$$

As with the previous analysis, the participants' individual slopes—averaged across the two repeating sequences—were included in the group level $t$-test for every ROI.

*Between the two repeating sequences and the unique sequences*

We tested whether the distance between $N$-th repetition of the repeating sequence $R_i$ and the twelve unique sequences $U_1, \ldots, U_{12}$ changed significantly across the experiment.

$$d_n = \mathbf{E}[distance(R_i^n, U_1), \ldots, distance(R_i^n, U_{12})].$$

*Within unique sequences*

We tested whether there was a change in pattern distances across successive presentations of individual unique sequences.

$$d_n = distance(U^n, U^{n-1}).$$

## Behavioural measures

Significant differences in behavioural measures across participants were evaluated with a t-test for dependent measures. We chose not to inverse-transform reaction time data following recent advice by Schramm and Rouder [87] (see also [88]).

## fMRI data acquisition and pre-processing

**Acquisition.** Participants were scanned at the Medical Research Council Cognition and Brain Sciences Unit (Cambridge, UK) on a 3T Siemens Prisma MRI scanner using a 32-channel head coil and simultaneous multi-slice data acquisition. Functional images were collected using 32 slices covering the whole brain (slice thickness 2 mm, in-plane resolution $2 \times 2$ mm) with acquisition time of 1.206 seconds, echo time of 30ms, and flip angle of 74 degrees. We used a multi-band factor of 2, partial Fourier 7/8, and applied no parallel imaging techniques

(partial Fourier only in the form of in-plane acceleration). In addition, high-resolution MPRAGE structural images were acquired at 1mm isotropic resolution. (See http://imaging. mrc-cbu.cam.ac.uk/imaging/ImagingSequences for detailed information). Each participant performed two scanning runs and 510 scans were acquired per run. The initial ten volumes from the run were discarded to allow for T1 equilibration effects. Stimulus presentation was controlled by PsychToolbox software [89]. The trials were rear projected onto a translucent screen outside the bore of the magnet and viewed via a mirror system attached to the head coil.

**Anatomical data pre-processing.** All fMRI data were pre-processed using *fMRIPprep* 1.1.7 [90, 91], which is based on *Nipype* 1.1.3 [92, 93]. The T1-weighted (T1w) image was corrected for intensity non-uniformity (INU) using `N4BiasFieldCorrection` ([94], ANTs 2.2.0), and used as T1w-reference throughout the workflow. The T1w-reference was then skull-stripped using `antsBrainExtraction.sh` (ANTs 2.2.0), using OASIS as target template. Brain surfaces were reconstructed using `recon-all` ([95], FreeSurfer 6.0.1), and the brain mask estimated previously was refined with a custom variation of the method to reconcile ANTs-derived and FreeSurfer-derived segmentations of the cortical grey-matter of Mindboggle [96]. Spatial normalisation to the ICBM 152 Nonlinear Asymmetrical template version 2009c [97] was performed through nonlinear registration with `antsRegistration` ([98], ANTs 2.2.0), using brain-extracted versions of both T1w volume and template. Brain tissue segmentation of cerebrospinal fluid (CSF), white-matter (WM) and grey-matter (GM) was performed on the brain-extracted T1w using `fast` ([99], FSL 5.0.9).

**Functional data pre-processing.** The BOLD reference volume was co-registered to the T1w reference using `bbregister` (FreeSurfer) using boundary-based registration [100]. Co-registration was configured with nine degrees of freedom to account for distortions remaining in the BOLD reference. Head-motion parameters with respect to the BOLD reference (transformation matrices and six corresponding rotation and translation parameters) were estimated using `mcflirt` ([101], FSL 5.0.9). The BOLD time-series were slice-time corrected using `3dTshift` from AFNI [102] package and then resampled onto their original, native space by applying a single, composite transform to correct for head motion and susceptibility distortions. Finally, the time-series were resampled to the MNI152 standard space (ICBM 152 Nonlinear Asymmetrical template version 2009c, [97]) with a single interpolation step including head-motion transformation, susceptibility distortion correction, and co-registrations to anatomical and template spaces. Volumetric resampling was performed using `antsApplyTransforms` (ANTs), configured with Lanczos interpolation to minimise the smoothing effects of other kernels [103]. Surface resamplings were performed using `mri_vol2surf` (FreeSurfer).

Three participants were excluded from the study because more than 10% of the acquired volumes had extreme inter-scan movements (defined as inter-scan movement which exceeded a translation threshold of 0.5mm, rotation threshold of 1.33 degrees and between-images difference threshold of 0.035 calculated by dividing the summed squared difference of consecutive images by the squared global mean).

## fMRI event regressors

To study sequence-based pattern similarity across all task phases we modelled the presentation, delay, and response phases of every trial (Fig 2A) as separate event regressors in the general linear model (GLM). We fitted a separate GLM for every event of interest by using an event-specific design matrix to obtain each event's estimate including a regressor for that event as well as another regressor for all other events (LS-S approach in Mumford et al. [104]).

Besides event regressors, we added six head motion movement parameters and additional scan-specific noise regressors to the GLM (see Functional data pre-processing above). The regressors were convolved with the canonical hemodynamic response (as defined by SPM12 analysis package) and passed through a high-pass filter (128 seconds) to remove low-frequency noise. This process generated parameter estimates (beta-values) representing every trial's task phases for every voxel. The beta values were then transformed to $t$-values to provide noise-normalised measure of voxel activity for the event.

We segmented each participants' grey matter voxels into anatomically defined regions of interest (ROI, $n = 74$). These regions were specified by the Destrieux [105] brain atlas and automatically identified and segmented for each participant using `mri_annotation2label` and `mri_label2vol` (FreeSurfer).

### Significance testing

We carried out the representational similarity analysis for every task phase (encoding, delay, response; $n = 3$) and ROI ($n = 74$ for RSA). To test whether the results were significantly different from chance across participants we used bootstrap sampling to create a null-distribution for every result and looked up the percentile for the observed result. We considered a result to be significant if it had a probability of $p < \alpha$ under the null distribution: this threshold $\alpha$ was derived by correcting an initial 5% threshold with the number of ROIs and task phases so that $\alpha = 0.05/74/3 \approx 10^{-4}$.

We next shuffled the sequence labels randomly to compute 1000 mean RSA correlation coefficients (Eq 23). To this permuted distribution we added the score obtained with the correct labelling. We then obtained the distribution of group-level (across participants) mean scores by randomly sampling mean scores (with replacement) from each participant's permuted distribution. The number of random samples for the group mean distribution was dependent on the significant probability threshold: we took $n = 10/\alpha$ samples so that the number of samples was always an order of magnitude greater than the required resolution for the group chance distribution. Next, we found the true group-level mean score's empirical probability based on its place in a rank ordering of this distribution.

### Supporting information

**S1 Text. Neural representation of novel sequences.**
(PDF)

**S2 Text. Item mixture model parameters.**
(PDF)

**S3 Text. Optimal chunking model estimation.**
(PDF)

**S4 Text. Associative learning of overlapping sequences.**
(PDF)

**S5 Text. Simulation of expected changes in pattern similarity.**
(PDF)

**S6 Text. Individual sequences used in the task.**
(PDF)

**S1 Fig. Anatomical coverage of MRI functional scans.**
(PDF)

## Acknowledgments

We would like to thank Jane Hall, Marta Correia, and Marius Mada for their assistance in setting up the experiments and collecting data.

## Author Contributions

**Conceptualization:** Kristjan Kalm.

**Formal analysis:** Kristjan Kalm.

**Funding acquisition:** Dennis Norris.

**Investigation:** Kristjan Kalm.

**Methodology:** Kristjan Kalm.

**Project administration:** Kristjan Kalm.

**Software:** Kristjan Kalm.

**Supervision:** Kristjan Kalm.

**Validation:** Kristjan Kalm.

**Visualization:** Kristjan Kalm.

**Writing – original draft:** Kristjan Kalm.

**Writing – review & editing:** Kristjan Kalm, Dennis Norris.

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
