## [Decision Letter · Decision Letter 0]

8 Feb 2021

Dear Kalm,

Thank you very much for submitting your manuscript "Sequence learning recodes cortical representations instead of strengthening initial ones" for consideration at PLOS Computational Biology. In light of the reviews (below this email), we would like to invite the resubmission of a significantly-revised version that takes into account the reviewers' comments.

My apologies for taking such a long time to come back to you. I had difficulties securing a third reviewer (blame the Christmas holidays), to the extent that I was still looking for one when the first two reviewers had submitted their evaluations. Since those reviews turned out to be highly detailed and consistent in their recommendation, I decided to move forward without a third reviewer.

As you will see, both reviewers were generally enthusiastic about your manuscript, but they also had a number of serious concerns that should be addressed. I will highlight the main issues here, but please see the full reviews below for more details and a range of smaller concerns that have been raised.

The first reviewer – an expert in cognitive modeling – found the manuscript difficult to understand, both in terms of its organization and the presentation and role of models in this study. I have a similar background as this reviewer and experienced the same problem. I believe that the various suggestions made by the reviewer can help to significantly improve the clarity of the modeling part of the paper. (Interestingly, the second reviewer – an expert in neuroimaging – indicated that the manuscript is well written and organized; hence, it seems that the manuscript is currently well adapted to a neuroimaging audience, but less so to a modeling audience.)

The main concern of the second reviewer is that inferences about the neural mechanisms are a bit surprising and possibly too strong due to the possibility of alternative explanations. The reviewer makes several suggestions to address this concern.

An additional comment that I have is that it could be useful to include a brief discussion about the possible role of long-term memory in these results. One question that I kept asking myself is whether part of the results may be explained as a consequence of a shift from working memory to long-term memory in trials with repeated sequences, while the novel sequences would always have to rely solely on working memory. I don’t know whether long-term memory processes should be seen as an alternative explanation or rather as a possible mechanism behind the recoding hypothesis. Some discussion on this would be helpful.

We cannot make any decision about publication until we have seen the revised manuscript and your response to the reviewers' comments. Your revised manuscript is also likely to be sent to reviewers for further evaluation.

Sincerely,

Ronald van den Berg

Associate Editor

PLOS Computational Biology

Samuel Gershman

Deputy Editor

PLOS Computational Biology

Reviewer's Responses to Questions

**Comments to the Authors:**

Reviewer #1: # Summary:

In this paper the authors compare an associate learning account with a recoding learning account for learning of short sequences (of length 4) using fMRI and representational similarity analysis (RSA). To this end, the predicted similarities across all items of four different representation models are created: an item-position model, an item-item model, an item mixture model, and an optimal chunking model. All of these are in some sense optimal models that produce their predictions without any free parameters (i.e., if a model has parameters, these are fixed to constants). The resulting similarity matrices are compared with the representational similarity matrix of the fMRI. The only prediction matrix that shows a significant relationship with the observed fMRI similarity is the optimal chunking model suggesting that only recoding explains the fMRI data. This results is further supported by a theoretical analysis (which shows that associative learning does not make much sense in the first place) and further fMRI data.

# Evaluation:

I really like the idea of the paper, the design of the study, the analysis, and the results. It also addresses a current research question in an active research area and does so using state-of-the-art methods, so I would like to see it published in a first class venue such as PLOS Comp Biol. However, I found the paper rather difficult to understand. One reason for this might be my lack of familiarity with the relevant fMRI/RSA literature, but I expect that with a better organisation of the manuscript could be made more accessible for a larger audience. In general I think there are some details missing from introduction and results that should be added to make the exposition clearer.

As someone with some experience in cognitive modelling, I had some problem understanding the role the models played here. To me, a model is usually something that has free parameters that are fitted to the data, but this is not the case here. All models make deterministic predictions a priori. I am not sure if I missed this in the manuscript, but it only became clear in the method section and was necessary in understanding the results. Maybe this could be presented clearer.

There are also some phrases that seem to be somewhat misleading about the role of the models. For example, line 175 reads: "The resulting RDM of n-gram distances was then fit with neural activity patterns using the RSA method." However, the "fitting" here is the calculation of the distance between two similarity matrices using rank correlations, which serves here as a measure of model adequacy, and not the more common procedure in which free parameters are adjusted to make the prediction of the model be nearer to the observed data. Similar phrases (i.e., estimated or fitted) can be found in lines 101, 161, and 252.

Likewise, Figure 4 refers to "Bayesian model comparison". In most contexts this phrase indicates that different models are compared against the data, but here it is only used to determine the structure of the the optimal chunking (i.e., recoding) model before any data is collected.

I generally fund the organisation of the results section somewhat unclear as it jumps between different type of results (i.e., the main RSA results and other fMRI results) and is organised across models. Maybe a clearer organisation in which results are presented one by one for each analysis (and not for each model) might be better. For example, the main result seems to be that only the similarity matrix of the optimal chunking/recoding model shows a significant relationship with the observed similarity matrix, but not for the other three accounts. However, the only graphical representation of this result is Figure 5 which only contrasts 2 models and not all 4.

I also found the additional prediction and analysis with respect to the noise activity the most difficult part of the analysis. Maybe this could be presented after the description of the main RSA results.

On the positive side, the implementation of the four sequence representation models appears to be done very thoroughly and I do not see any large issues. The one thing I think would be interesting to discuss if there is the possibility that additional free parameters (e.g., learning rates or decay) could alter any of the predictions of the models substantially. I do not suggest that such a model should be analysed (it would also be unclear how), but rather whether the deterministic predictions are really the only plausible way associations models could create similarities.

I list a few more comments below that I had while reading the manuscript:

- The description of the results is difficult to understand without reading the method section. Perhaps a few more information about the task could be given. For example, I had the following two questions: What was the task of the participants? Does the task consist of two parts (initial learning phase without fMRI followed by final part with fMRI) or just one part?

- A response time difference of 0.018 seconds (i.e., 18 milliseconds) does not seem to impressive. What is the average RT?

- Fig. 1: I do not know the term "multinomial matrix representation". I understand the figure, but the term is at least uncommon to me.

- Fig 2B. Either there is an error in the id column (trial 5 seems to have the wrong id) or I am misunderstanding what it means.

- Fig. 3 (left): What is the distance measure used to derive the predicted distance between two sequences?

- line 156: "recoded inferred": only one of the two words is necessary

- Fig. 4: What does "Repeating 2" under "Trial 3" mean?

- lines 373 - 377: It is a bit unclear if the randomisation guarantees that a repeating sequence would always occur in any series of three trials - with the consequence that if one hasn't seen a repeating sequence in the last two trials the next trial would be a repeating sequence with certainty - or whether a repeating sequence would only certainly follow after four non-repeating sequences.

- Line 365: I think it would be good to alert the reader here that the reason why there are only 14 individual sequences is revealed later on. I spent too long wondering why 4! would result in only 14 sequences.

Reviewer #2: The present study investigates the neural basis of sequence learning. In particular, two models of sequence learning, associative and recoding (chunking), are formulated and their fit to fMRI data is compared. Throughout several areas in the dorsal visual stream, the recoding model shows a superior fit to the fMRI data. Additionally, the authors provide model evidence that the associative learning model suffers from significant interference when many possible sequences are learned. Combined, the authors conclude that only the recoding model can account for visual sequence learning.

The manuscript is well written and organized. The models appear well specified and the fMRI data analysis is well conducted, using state of the art tools and methods. Overall, the analyses and their description are of high quality. I also appreciate the authors’ commitment to open science by sharing all data and analysis code. The major strength of the present study over most previous ones is the explicit formulation of the learning models, which promises to yield significant additional insight into the neural mechanisms underlying human sequence learning. However, I do have some concerns with the design and interpretation of the results, questioning whether the study truly achieves to provide this additional insight.

Major points

1. To my knowledge, the neural consequences of visual sequence learning (or visual statistical learning) are commonly reported in primary and ventral visual areas, as also reflected by several papers cited by the authors (e.g. [31-33] in the manuscript; also see de Lange et al., Trends Cogn Sci 2018 for a review). Another line of evidence suggests the involvement of subcortical structures, such as hippocampus, in statistical learning and positional codes (Hindy et al. 2016; Hsieh et al. 2014). Finally, frontal areas may also encode sequence information in serial order tasks (Berdyyeva and Olson, 2010). Given the prevalence of studies pointing towards sensory, frontal and subcortical areas as key nodes involved in sequence learning, I was surprised by the results in the present manuscript being located exclusively in dorsal visual stream regions (largely parietal cortex; Fig. 5). While the authors state that they want to address the underlying computational mechanism rather than where in the brain effects of learning are found (lines 54-56), I nonetheless believe that the unexpected localization of the results does at least warrants critical evaluation and discussion, also in the context of previous literature.

Indeed, the motivation for the present study is framed in the context of previous work on (visual) sequence/statistical learning and aims to advance our understanding by investigating the underlying computation rather than only a univariate difference in response magnitude – a goal I fully support. However, if the observed neural effects drastically differ in localization, is this goal really achieved in the present study? The authors compare their results to several studies showing modulations in early visual and ventral visual stream areas (e.g. line 195), yet report univariate and RSA results exclusively in dorsal visual areas. Without a convincing explanation for this discrepancy, and why it should not be of concern, it would seem to me that the present results do not provide novel insight into the specific mechanism that was cited to motivate their study.

2. Relatedly, the specifics of the experiment design and the cortical areas reported by the authors may in fact suggest an alternative explanation for the results. In the task performed by the participants, the recall of stimulus order, responding to novel compared to learned sequence is more difficult. Accordingly, also behavioral results show slower responses to the more difficult to recall novel sequences. Arguably, trials with novel sequences require more attention, likely over a longer duration, to be devoted to the stimuli. The univariate fMRI results are in line with this interpretation, by showing enhanced neural responses to novel compared to learned sequences; i.e., possibly an enhanced neural responses to novel sequences due to attention. Moreover, there appears to be an appreciable overlap of the results reported in Fig. 5, with posterior parts of the default mode and attention related networks. Combined this evidence suggests an interpretation stressing the role of attention allocation and task difficulty as underlying (at least partially) the observed effects. Therefore, to me it appears unclear what neural mechanism is reflected in the present results. This is a notable limitation, which should be critically evaluated by the authors, or if possible be addressed with additional analyses – although, unfortunately I do currently not see how the existing data could directly answer this concern.

3. The critical question is then whether differences in attention or related effects, brought about by differences in task difficulty and task demands between the conditions of interest, could also account for (apparent) changes in neural representations; the RSA results at the heart of the manuscript. To me this seems possible, albeit less straight forward than for the univariate results. Nonetheless, one can speculate.

If I understand correctly, the authors constrained sequence selection (maximized differences between sequences) and pseudo-randomized the order of trials (2 learned, 1 novel sequence for every set of 3 trials). This maximization of differences between sequences and 2nd order regularity across trials probably makes it possible for participants to rapidly and with high probably detect learned sequences after the first stimulus of a sequence is presented. I believe that the third sequence in a set of three trials can even be perfectly predicted before stimulus onset, if 1 learned and 1 novel trial have been seen so far. Even without this 2nd order regularity, learned sequences can be detected rapidly. That is, if the first stimulus belongs to one of the two learned sequence chances are very high that the sequence is the learned sequence; because only 1/3 of sequences are novel sequences and of those only 1/4 start with any one particular stimulus, while on the other hand 2/3 of trials are learned sequences of which each starts with a different stimulus. Thus, for learned sequences participants can rapidly disengage attention after recognizing the specific learned sequence as e.g. “sequence 1”, possibly already upon seeing the first stimulus in a sequence. Then participants can simply recall the required responses before preparing to perform the associated action. Note that no attention to subsequent stimuli is necessary in most cases. On the other hand, novel sequences require sustained attention to the stimulus sequence itself, do not allow observers to preemptively disengage, demand storing the associated stimuli in working memory, prepare a novel motor plan, etc. In other words, many aspects differ in how observers may respond to learned compared to novel sequences. Importantly, many of these differences do not appear to be related to how learning changes the representation of a sequence itself or the representation of stimuli in a sequence, but rather are secondary consequences of having learned sequences in this task, and of making use of what has been learned.

It seems possible that one of these differences, or a combination thereof, may result in a better fit of the recoding model compared to the associative learning model. Consider that a learned sequence can be recognized rapidly (e.g. as “sequence 1”) and attention disengaged from the stimuli, only requiring the participant to recall the motor sequence to be performed. The associated representation of this recognition and the secondary processes outlined above would be highly unlikely to resemble the predictions of the associative learning model, but rather resemble a new “recoded” representation. However, this does not seem to me to indicate that the representation of the sequence has necessarily been recoded, but another secondary process may mask as a recoded sequence representation.

Even so, it is interesting how these secondary consequences modulate neural responses – i.e., I do not want to argue that the results are uninteresting. Rather, the problem is again that we do not seem to know which mechanism, modulation, strategy, etc. is underlying the present results. In a study that aims to narrow down the precise neural mechanism underlying sequence learning this uncertainty about which process may underpin the observed results is, in my opinion, a crucial limitation worth discussing.

Minor points

1. Related to the points above, one may also wonder whether the discrepancy in results compared to previous studies could also be a consequence of differences in stimulus duration. Compare for example the fast-paced statistical learning in [31-33] (stimulus presentation <=500ms), which resulted in reduced sensory responses to learned sequences in early and ventral visual areas, with the slow paced presentation (2.4s) in the present study. Pacing could thus be an important characteristic relevant for determining which coding strategy is employed. If this is the case, it would limit the generalization of the conclusions we can draw from the present study.

2. In lines 118-124, the authors explain the rationale of the test of the associative learning model. I understand the incentive to use RSA here as well. However, it seems to me that a decoding approach (e.g. SVM), comparing decoding accuracy of novel compared to learned sequences, may have provided a more powerful approach, assuming appropriate training data would have been acquired. It might be worth explaining why RSA would be preferable over a decoding analysis (or at least equally suited) to assess predictions of the associative learning model. Otherwise, it may appear to the reader that the study design and analysis procedure is optimized for the recoding model.

3. The authors conclude “humans follow an optimal sequence learning strategy and recode initial sequence representations into more efficient chunks” (line 314-315). What is meant here by “optimal”? I am not sure I quite see how the data support that an optimal strategy is necessarily used, but only that recoding fits the current data better than the associative learning model and the null model. I do understand that practical limitations constrain which models can be implemented, but (at least in my understanding) it has not been tested whether an optimal learning strategy is indeed used.

4. Lines 123-124 first introduce the dorsal visual stream as a ROI. This choice of ROI should be supported or introduced in some fashion.

5. Lines 126-127 state that “We found no evidence for the first associative learning prediction: novel and repeating sequences were not encoded similarly in any of the brain regions”. Would it be possible to clarify which results the authors are referring to here?

6. In addition to the lower and upper noise ceiling for the model results, it might be helpful to mention the variance explained by the models.

7. Important additional MR sequence parameters, such as the multi-band factor, etc., would be a good addition to the Methods section (around line 782). This would better allow the reader to evaluate potential shortcomings of the utilized sequence.

8. Has the MR sequence been evaluated for slice leakage?

9. Was mean scaling performed as part of the univariate fMRI analysis pipeline?

10. Several figures (e.g. Fig. 5, Fig 9) would benefit from labels on the y-axis and the colorbar.

11. The authors mention that in ¼ of trials the recall phase was omitted. Was this data used in any specific manner? What was the intention behind this design choice? The stated rationale, “to ensure a sufficient degree of decorrelation between the estimates of the BOLD signal for the delay and recall phases” would have been possible to achieve by using a variable duration for the delay window.

12. I could not find the fMRI dataset under the link provided in the manuscript. I assume that the authors will publish the dataset upon publication?

13. I can confirm that I could access the code shared on gitlab. The code seems well written and documented. Sharing code in a well readable format is much appreciated. That said, I did not have time to adequately review the code or test it, also due to the problems with data access.

**Have all data underlying the figures and results presented in the manuscript been provided?**

Reviewer #1: Yes

Reviewer #2: **No: **I could not access the MRI dataset using the link in the manuscript, and I did not notice any separate numerical data underlying the figures. However, this could also have been an oversight/problem on my part.

PLOS authors have the option to publish the peer review history of their article (what does this mean?). If published, this will include your full peer review and any attached files.

Reviewer #1: No

Reviewer #2: No
---

## [Decision Letter · Decision Letter 1]

16 Apr 2021

Dear Kalm,

We are pleased to inform you that your manuscript 'Sequence learning recodes cortical representations instead of strengthening initial ones' has been provisionally accepted for publication in PLOS Computational Biology.

Best regards,

Ronald van den Berg

Associate Editor

PLOS Computational Biology

Samuel Gershman

Deputy Editor

PLOS Computational Biology

Reviewer's Responses to Questions

**Comments to the Authors:**

Reviewer #1: The revision does a good job in making the manuscript clearer and I am happy to see it published. I only have a few comments the authors might want to consider:

- lines 102 to 105: I think it would be helpful to clarify here that the prediction is parameter-free or does not depend on participants' responses. This is the important part where a misunderstanding of what is meant with model here can make it difficult to understand the results.

- I am not 100% convinced the response to point 1.14 is sufficient. The manuscript now reads (ll. 492): "To keep the relative probability of repeating and unique sequences fixed throughout the experiment we pseudo-randomised the order of trials so that on the average there was a single unique sequence and two repeating sequences in three consecutive trials (Fig 2B)." The phrase "on average" suggests that it does not hold for any set of three trials. Maybe it would be clearer to describe the randomisation procedure. E.g.: The trial list was divided in sub-blocks of length 3 each of which contained exactly one repeated sequence with position randomly determined.

- Figure 3 note: Two opening parentheses in new part, but only one closing parenthesis.

- line 200 (likewise in line 243): "p > 10^-3" seems a surprisingly low threshold to declare an effect as absent. I understand the logic (ll. 942), but it still feels inappropriate.

Reviewer #2: The authors well addressed the concerns and comments raised by the other reviewers and myself. Particularly, the major comments have been discussed thoroughly, which is much appreciated. My questions about the anatomical localization of the MRI results have been satisfactorily addressed. In my opinion the associated adjustments to the manuscript, and the discussion of alternative learning mechanisms, make the manuscript more accessible and better places the results in the context of the wider literature. Explaining the rationale and limitations of the fMRI slice position and coverage further clarifies the localization of the results and absence of results in other areas. In short, I believe that the manuscript is publishable in its current form. That said, I would briefly like to mention a few remaining minor comments for your consideration.

Minor comments

1. Line 64: “Specifically, the code for of sequences changed from […]” accidentally says “for of”

2. I appreciate the adjustments to the Discussion. However, the new version of the discussion occasionally seem to lack focus; e.g. the paragraphs of lines 300, 319, 347 all deal with potential hippocampus related learning mechanisms in one way or another, but are separated by other parts of discussion. Although this might not be strictly necessary, it would be helpful if it were possible to slightly streamline the Discussion.

3. In their reply the authors state that: “In our view the identification-and-disengage mechanism is only possible using a latent variable that maps onto a longer sequence. This model can be called pattern completion or chunking — in our manuscript we use the neutral term recoding.” I appreciate that the authors provided an elaborate explanation how they believe this alternative account is covered in their model. I do also agree with the authors that such an “identification-and-disengage” mechanism may indeed require recoding. However, it does still raise the question what the changes in fMRI patterns reflect – is it necessarily the representation of the recoded sequence or a (secondary) consequence thereof; e.g. related to attention. To be clear, the manuscript answers the question whether recoding or associative learning better match the fMRI patterns, thus my comment does not question whether the authors achieved the goal of the study.

**Have the authors made all data and (if applicable) computational code underlying the findings in their manuscript fully available?**

Reviewer #1: None

Reviewer #2: **No: **However the authors stated that: "[...] once we have received conditional acceptance for our manuscript we can link the data immediately. This can be verified by the editor and the reviewers before making the manuscript public on the PLOS website."

I understand the explanation why the dataset is not available during review and I do realize that such SOPs are probably decided on an institute level. However, for review transparency it would be great if datasets could be made available for reviewers (e.g. with a personalized reviewer link) for future manuscripts.

PLOS authors have the option to publish the peer review history of their article (what does this mean?). If published, this will include your full peer review and any attached files.

Reviewer #1: No

Reviewer #2: No

---

## [Editor Report · Acceptance letter]

19 May 2021

PCOMPBIOL-D-20-02080R1 

Sequence learning recodes cortical representations instead of strengthening initial ones

Dear Dr Kalm,

I am pleased to inform you that your manuscript has been formally accepted for publication in PLOS Computational Biology. Your manuscript is now with our production department and you will be notified of the publication date in due course.

With kind regards,

Olena Szabo
